# Genome editing of an African elite rice variety confers resistance against endemic and emerging *Xanthomonas oryzae* pv. *oryzae* strains

Van Schepler-Luu[1,2†], Coline Sciallano[3†], Melissa Stiebner[1†], Chonghui Ji[4†], Gabriel Boulard[3], Amadou Diallo[3], Florence Auguy[3], Si Nian Char[4], Yugander Arra[1], Kyrylo Schenstnyi[1], Marcel Buchholzer[1], Eliza PI Loo[1], Atugonza L Bilaro[5], David Lihepanyama[5], Mohammed Mkuya[6], Rosemary Murori[7], Ricardo Oliva[2‡], Sebastien Cunnac[3], Bing Yang[4,8], Boris Szurek[3]*, Wolf B Frommer[1,9]*

[1]Institute for Molecular Physiology, Heinrich Heine University Düsseldorf, Düsseldorf, Germany; [2]International Rice Research Institute, Los Baños, Philippines; [3]Plant Health Institute of Montpellier (PHIM), Université Montpellier, IRD, CIRAD, INRAE, Institut Agro, Montpellier, France; [4]Division of Plant Science and Technology, Bond Life Sciences Center, University of Missouri, Columbia, United States; [5]Tanzania Agricultural Research Institute (TARI)-Uyole Centre, Mbeya, United Republic of Tanzania; [6]International Rice Research Institute, Eastern and Southern Africa Region, Nairobi, Kenya; [7]International Rice Research Institute (IRRI), Africa Regional Office, Nairobi, Kenya; [8]Donald Danforth Plant Science Center, St. Louis, United States; [9]Institute for Transformative Biomolecules, ITbM, Nagoya University, Nagoya, Japan

*For correspondence:
boris.szurek@ird.fr (BS);
frommew@hhu.de (WBF)

†These authors contributed equally to this work

Present address: ‡new address: World Vegetable Center, Shanhua, Tainan, Taiwan

**Abstract** Bacterial leaf blight (BB) of rice, caused by *Xanthomonas oryzae* pv. *oryzae* (*Xoo*), threatens global food security and the livelihood of small-scale rice producers. Analyses of *Xoo* collections from Asia, Africa and the Americas demonstrated complete continental segregation, despite robust global rice trade. Here, we report unprecedented BB outbreaks in Tanzania. The causative strains, unlike endemic African *Xoo*, carry Asian-type TAL effectors targeting the sucrose transporter *SWEET11a* and iTALes suppressing *Xa1*. Phylogenomics clustered these strains with *Xoo* from Southern-China. African rice varieties do not carry effective resistance. To protect African rice production against this emerging threat, we developed a hybrid CRISPR-Cas9/Cpf1 system to edit all known TALe-binding elements in three *SWEET* promoters of the East African elite variety Komboka. The edited lines show broad-spectrum resistance against Asian and African strains of *Xoo*, including strains recently discovered in Tanzania. The strategy could help to protect global rice crops from BB pandemics.

## Editor's evaluation

This valuable study shows that new, virulent genotypes of Xanthomonas oryze pv. oryzae, that are similar to strains present in east Asia, cause outbreaks of bacterial blight of rice in Tanzania. The authors' use of CRISPR-based gene editing on multiple pathogen targets in an elite African rice variety to create lines resistant to both endemic and emerging pathogen strains in Africa makes for a compelling contribution to meet this alarming development.

## Introduction

Rice is one of the most important staple foods for developing countries in Asia and Africa (*Odongo et al., 2021*). African consumers increasingly replace traditional staples such as sorghum, millet, and maize with rice. Today, African farmers, 90% of which are small-scale food producers with <1 ha of land (*Pandey et al., 2010*), produce ~60% of local rice demand. The demand will likely increase with the expected doubling of the population until 2050. Productivity is often hampered by diseases, such as Bacterial Leaf Blight (BB), Rice Yellow Mottle Virus (RYMV), and Rice Blast (RB) (*Jiang et al., 2020*; *Longue et al., 2018*; *Mutiga et al., 2021*). Breeding high-yielding varieties with resistance to these diseases will be an important factor needed for food security in Africa. BB, caused by the bacterium *Xanthomonas oryzae* pv. *oryzae* (*Xoo*), is a devastating rice disease in many rice-growing countries. *Xoo* comprises a wide spectrum of pathovars with diverse virulence mechanisms. In rice, resistance (*R*) genes for BB have been identified and are used extensively to breed resistant varieties. However, resistance based on single *R* genes was rapidly overcome by new pathovars (*Vera Cruz et al., 2000*). Regular monitoring of the virulence of current *Xoo* populations on a collection of rice tester lines carrying single or combinations of *R* genes has provided effective guidance for stacking different *R* genes to obtain broad-spectrum resistance in Asian rice varieties (*Arra et al., 2017*; *Arra et al., 2018*).

In Africa, BB was first reported in Mali and Cameroon in the late 1970s, and later in several other West African countries (*Buddenhagen et al., 1979*; *Verdier et al., 2012b*). Recently, East African countries also reported BB epidemics (*Duku et al., 2016*). While information on the spread and severity of BB in Africa is scarce, BB is not yet considered a major threat in Africa; nevertheless, BB is an established disease. The situation is unstable, because climate change affects disease spread, and damage is expected to become more severe in the future due to climate change (*Amos, 2013*; *Laha et al., 2016*; *Sere and Ouedraogo, 2005*; *Verdier et al., 2012b*). Increased demand and the need to improve food security make it essential to generate broad-spectrum and durable resistance in local varieties specifically for Africa.

Key aspects of the mechanisms underlying BB disease have been elucidated and provide an efficient roadmap for resistance breeding. *Xoo* strains secrete a suite of Transcription Activator-Like effectors (TALes) into rice xylem parenchyma cells via type III secretion systems. Once inside the host cell, TALes are targeted to the nucleus, where they bind to effector binding elements (EBEs) in the promoters of specific host genes via a unique domain of tandemly arranged 34 amino-acid long repeats. TALes function as transcriptional activators that trigger ectopic induction of target genes (*Richter et al., 2014*). Several *Xoo* TALes induce one or several host SWEET sucrose uniporter genes (*SWEET11a, 13* and *14*), presumably causing sucrose release from the xylem parenchyma into the apoplasm at the sites where the bacteria reside and reproduce (note that due to the discovery of a previously misannotated sucrose transporting paralog, SWEET11a was renamed formerly SWEET11 *Wu et al., 2022*). Sucrose is consumed by the bacteria, resulting in effective reproduction (*Sadoine et al., 2021*). Allelic EBE variants that cannot be recognized by TALes function as recessive gene-for-gene resistance genes. Without SWEET induction, *Xoo* is not a potent pathogen. To date, six target EBEs in *SWEET* promoters have been identified in a comprehensive *Xoo* collection. Notably, Asian and African *Xoo* strains are phylogenetically distinct and use distinct TALes to target *SWEETs* (*Oliva et al., 2019*; *Tran et al., 2018*). African *Xoo* strains exclusively use TalC and TalF, which both target *SWEET14*, while Asian strains encode PthXo1, PthXo2 (variants A, B, C), PthXo3 and AvrXa7, which target *SWEET11a, 13,* and *14,* respectively (*Eom et al., 2019*; *Oliva et al., 2019*). American *Xoo* strains lack TALes and hence are poorly virulent (*Verdier et al., 2012a*). Based on the information of the TALe compendium and the *SWEET* target sites, broad-spectrum resistance to Asian strains was successfully introduced into *Oryza sativa* ssp. *japonica* cv. Kitaake and the spp. *indica* varieties IR64 and Ciherang-Sub1 by genome editing the EBEs in all three *SWEET* promoters (*Eom et al., 2019*; *Oliva et al., 2019*). The edited lines may prove to be valuable breeding materials that could benefit small-scale producers in Asia. To prepare for emerging strains with novel virulence mechanisms, the diagnostic SWEET$^R$ kit was developed, enabling effective characterization of causative strains and their SWEET-based disease mechanisms, and identification of suitable resistance strategies, such as the best possible *SWEET$^R$*-gene combinations for deployment in the target country (*Eom et al., 2019*).

First observations of BB in Africa were made as early as 1979 in Mali and Cameroon (*Buddenhagen et al., 1979*). Since then, the disease has been reported mostly in West Africa, including Senegal, Benin, Burkina Faso, Ivory Coast, Mali, and Niger (*Afolabi et al., 2016*; *Gonzalez et al., 2007*; *Tall*

*et al., 2020*; *Tekete et al., 2020*). More recently, BB was also reported in the East African countries Uganda and Tanzania (*Duku et al., 2016*; *Oliva et al., 2019*). However, at present, little information on the genetic diversity and dynamics of *Xoo* populations is available for Africa. While pathogenic strains have sporadically been isolated from rice plants and characterized, there has been no systematic analysis of *Xoo* strains and BB disease severity across Africa. Moreover, diverse rice varieties are used across Africa, and there is only limited information on the acreage used for different varieties. These factors make it challenging to breed durably resistant lines for Africa. Numerous international partnerships have introduced varieties and cultivation techniques to major rice production areas in different countries in Africa. For instance, in 1975, one of the first irrigated perimeters with rice research labs and a training farm was developed in Dakawa, a major rice production zone in Tanzania, as a result of a multi-year partnership with North Korea (*van et al., 2022*). National (NAFCO) and international institutions (USAID, Cornell University) remain active in Dakawa to build irrigation schemes and implement *The System of Rice Intensification* (SRI). In 2011, a collaboration with China resulted in 50 ha of irrigated fields managed by local farmers. A Technology Demonstration Center that trains local farmers and technicians and grows Chinese varieties, including hybrid rice has been established (*Makundi, 2017*).

Recent disease surveys in Tanzania, described here, identified an outbreak of BB in 2019 that has spread in subsequent years. We show that the causative strains have features that distinguish them from endemic African *Xoo* strains, and unravelled the virulence mechanisms that make them a major threat to African rice production. Recently, Kenya and Nigeria exempted SDN-1 genome-edited crops generated using site-directed nuclease without a transgene (SDN-1) from GMO regulations (*Buchholzer and Frommer, 2023*). This exemption provides a regulatory framework for the introduction of genome-edited, BB-resistant rice into African countries such as Kenya and Nigeria. As a first step towards the generation of BB-resistant rice cultivars for small-scale rice producers in these countries, we used an optimized transformation protocol to edit susceptibility genes in the rice cultivar Komboka, which is popular in East African countries (*Luu et al., 2020*). Komboka (IRO5N-221) is a relatively new high-yielding, semi-aromatic rice variety jointly developed by IRRI (International Rice Research Institute) and KALRO (Kenya Agricultural & Livestock Research Organization; BBSRC Varieties Kenya.pdf) that grows to an average height of 110cm and has a yield potential of~7 tons/ha, - almost twice that of Basmati, another popular variety planted in Kenya and Rwanda (The Star 2020). Komboka plants mature in~115days, respond well to fertilization, and are suitable for irrigated lowland cultivation in Africa (*Kitilu et al., 2019*). Komboka had been classified as moderately resistant to BB, RYMV, and RB; however, detailed resistance profiles for specific strains or races are not available, rendering this information of little value for breeding resistance in the context of specific pathovars present locally. Here, BB-resistance and susceptibility genes of Komboka were analyzed, and a combination of Cas9, Cpf1 and multiplexed gRNAs was used to edit all known EBEs in the *SWEET* genes, resulting in broad-spectrum resistance to not only previously known African and Asian strains, but also the newly introduced *Xoo* strains identified from the outbreak in Tanzania.

## Results
### Identification of an unprecedented disease outbreak in Tanzania

Local breeders reported that although BB was found in Tanzania, it was not considered a major disease due to low incidence and low severity up until 2019 (RM, pers. comm.). Here, we report on an unprecedented BB outbreak identified in the irrigated schemes of Dakawa and Lukenge in 2019 and 2021, respectively (*Figure 1*; *Figure 1—figure supplement 1*, *Figure 1—figure supplement 2*). The two sites are ~60 km apart in the Morogoro region, an area that for decades has been a center of partnership initiatives that aim at increasing rice production (*Makundi, 2017*). We performed disease surveys in this area in 2019 and detected a severe outbreak in Dakawa on TXD 306 (SARO-5), a rice variety popular in irrigated ecologies in Tanzania (*Figure 1—figure supplement 1*). Later on, in 2021 in Lukenge, several fields also showed severe signs of infections (*Figure 1*; *Figure 1—figure supplement 2*). Leaves collected in 2019 and 2021 with typical BB symptoms were processed, resulting in the isolation of seven strains from Dakawa and 106 from Lukenge, all validated as *Xoo* using diagnostic primers (*Lang et al., 2010*; *Table 1*, *Supplementary file 1a*). In 2022, we surveyed BB on a larger scale and collected over 600 leaf samples from plants with typical BB symptoms from 37 fields

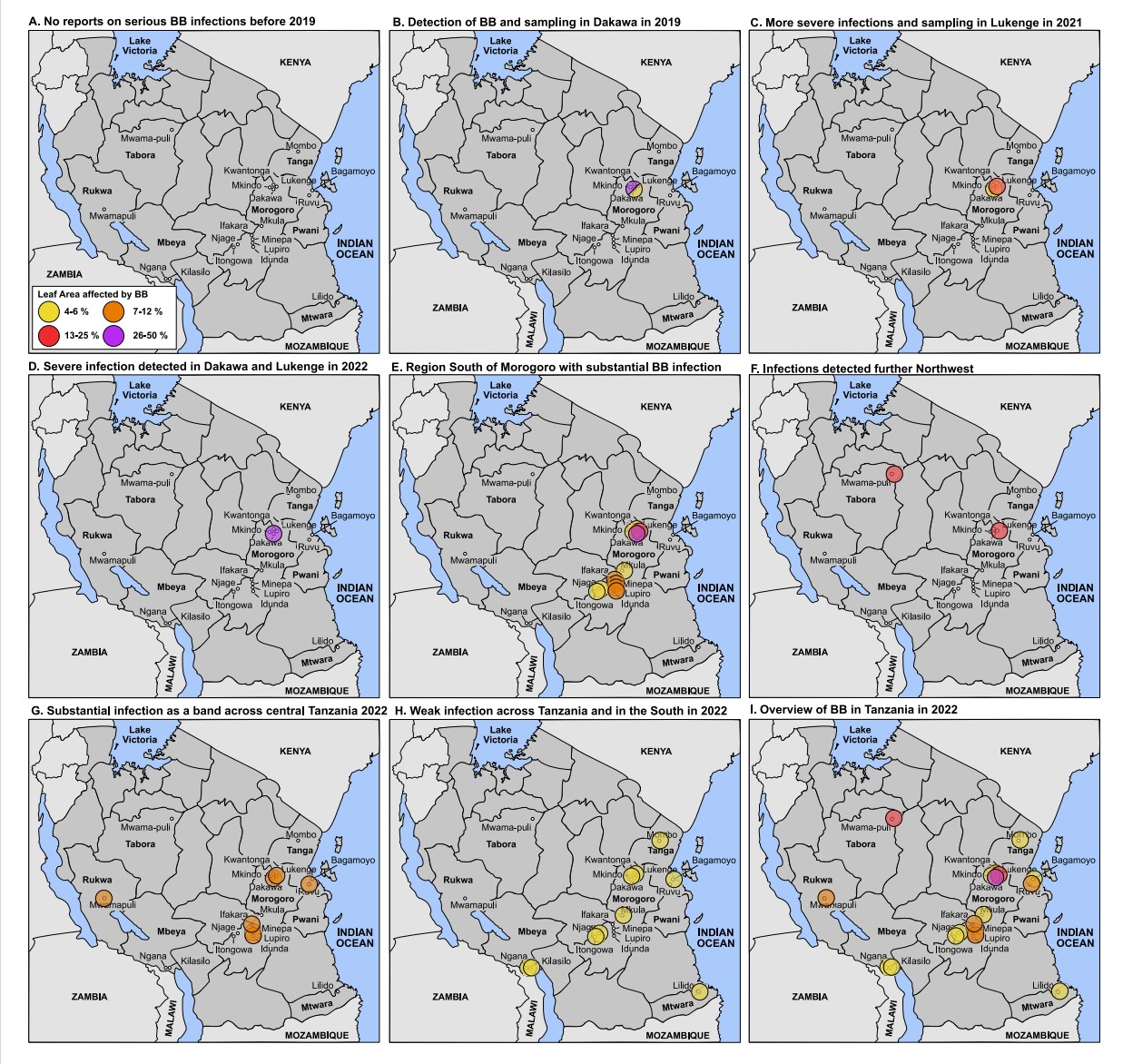

**Figure 1.** Detection of an outbreak and survey of BB in Tanzania. (**A**) No reports of serious BB infections before 2019. (**B**) Detection in Dakawa; in 2019, (**C**) in Lukenge in 2021, (**D–I**) survey results from 2022 at different degrees of severity. Note that sampling and scoring occurred initially in the Morogoro region in Dakawa in 2019 and in Lukenge 2021, (no info on 2020 due to Covid-19); scoring and sampling was expanded to additional fields across Tanzania in 2022. Due to sampling bias early in the outbreak, data should be interpreted with caution. One may project that BB caused by the introduced strains will manifest in Tanzania and could spread to neighboring countries. Leaf area affected: yellow 4–6%, orange 7–12%, red 13–25% and purple 26–50%.

The online version of this article includes the following figure supplement(s) for figure 1:

**Figure supplement 1.** Photos of infected fields showing severe infection of the field in Dakawa, Morogoro, Tanzania, in 2019.

**Figure supplement 2.** Photos of infected fields showing infection of the field in Lukenge, Morogoro, Tanzania, in 2021.

in six different regions in Tanzania. The results of the survey show that BB has spread across Tanzania (*Figure 1*; *Supplementary file 1b*). Multiple Locus Variable Number of Tandem Repeat Analyses and/ or whole genome SNP analysis is in progress and is expected to provide new insights into the genetic structure of the *Xoo* population in Tanzania. A careful analysis of yield losses has not yet been possible. Out of six fields surveyed in Dakawa in 2019, one was very severely infected (*Figure 1—figure supplement 1*). The yield loss in this field was estimated to be >60%. One of the fields surveyed in Lukenge in 2021 had an estimated yield loss of 10–15%. In 2022, several Dakawa fields had estimated yield losses

**Table 1.** Efficiency of resistance genes in IRBB rice lines towards a diversity panel of 26 African Xoo strains.

Quantitative scores of lesion length produced upon leaf-clip inoculation of a diversity panel consisting of 26 Xanthomonas oryzae pv. oryzae strains from West and East Africa on IRBB near-isogenic-lines (NILs) carrying different Xanthomonas resistance genes (Xa) and on the variety Azucena used as susceptible control (IRBB1: Xa1; IRBB3: Xa3; IRBB4: Xa4; IRBB5: xa5; IRBB7: Xa7; IRBB23: Xa23; IR24: recipient parent). Resistance or susceptibility of rice plants to Xoo was determined based on lesion length measured 14 days after inoculation: Resistant (R) <5 cm; moderately resistant (MR)=5–10 cm; moderately susceptible (MS)=10–15 cm; and susceptible (S) >15 cm.

| Strain name | CIX code | Country | Azucena | IR24 | IRBB1 | IRBB3 | IRBB4 | IRBB5 | IRBB7 | IRBB23 | Komboka |
|---|---|---|---|---|---|---|---|---|---|---|---|
| NatiPark | 607 | Benin | S | R | R | R | R | R | R | R | R |
| Karfiguela13 | 705 | Burkina Faso | S | R | R | R | R | R | R | R | R |
| N2-4 | 4482 | Niger | S | R | R | R | R | R | R | R | R |
| Tanguieta3 | 609 | Benin | S | MR | R | MR | R | R | MR | R | R |
| Toula20 | 629 | Niger | S | MR | R | MR | R | R | MR | MR | R |
| BAI250 | 4127 | Burkina Faso | S | MR | R | MR | MR | R | R | MR | R |
| S62-2-22 | 2374 | Senegal | S | MR | R | MS | R | R | MR | R | R |
| CII-1 | 4083 | Ivory Coast | S | MS | MR | MS | R | R | MR | MR | R |
| CII-2 | 1042 | Ivory Coast | S | MS | MR | MS | MR | R | MR | MR | R |
| NAI9 | 2787 | Niger | S | MS | MR | MS | MR | MR | MS | S | S |
| AXO1947 | 1917 | Cameroon | S | S | R | S | R | R | MS | MS | R |
| MAI145 | 894 | Mali | S | S | R | S | MR | MS | S | MR | R |
| BAI3 | 4092 | Burkina Faso | S | MS | MR | S | MR | MR | MS | S | R |
| CFBP1948 | 2801 | Cameroon | S | MS | MR | S | MR | MR | S | S | R |
| NAI5 | 4099 | Niger | S | S | MR | S | MR | MR | S | S | R |
| MAI73 | 4079 | Mali | S | S | R | S | MS | MR | S | MR | R |
| S82-4-3 | 2976 | Senegal | S | MS | MR | S | MS | MR | MS | S | R |
| MAI132 | 4517 | Mali | S | S | R | S | S | MS | S | MR | R |

*Table 1 continued on next page*

*Table 1 continued*

| Strain name | CIX code | Country | Azucena | IR24 | IRBB1 | IRBB3 | IRBB4 | IRBB5 | IRBB7 | IRBB23 | Komboka |
|---|---|---|---|---|---|---|---|---|---|---|---|---|
| iTzDak19-1 | 4457 | Tanzania | S | S | S | S | S | S | S | R | S |
| iTzDak19-2 | 4458 | Tanzania | S | S | S | S | S | S | S | R | S |
| iTzDak19-3 | 4462 | Tanzania | S | S | S | S | S | S | S | R | S |
| iTzLuk21-1 | 4506 | Tanzania | nd | nd | nd | nd | nd | nd | nd | nd | nd |
| iTzLuk21-2 | 4509 | Tanzania | nd | nd | nd | nd | nd | nd | nd | nd | nd |
| iTzLuk21-3 | 4505 | Tanzania | nd | nd | nd | nd | nd | nd | nd | nd | S |
| iTzLuk21-4 | 4507 | Tanzania | nd | nd | nd | nd | nd | nd | nd | nd | S |
| iTzLuk21-5 | 4508 | Tanzania | nd | nd | nd | nd | nd | nd | nd | nd | nd |

n.d. not determined.

of 5–10%, others <5%. Yield loss at other sites such as Igunga was estimated to 10%, in Mombo <5%. Note that estimates of severity and yield losses are snapshots, and careful and systematic multiyear analyses will be required for a reliable evaluation of severity, spread and yield losses.

To systematically analyze the newly isolated strains from Tanzania, and to compare them to the broader African *Xoo* landscape, we used 8 strains isolated from Dakawa in 2019 and Lukenge in 2021, as well as 18 representative strains from a collection of 833 strains sampled from rice fields in nine other African countries between 2003 and 2021. These endemic African strains had previously been identified as *Xoo* based on molecular diagnostic and pathogenicity assays (BS, unpubl. results;

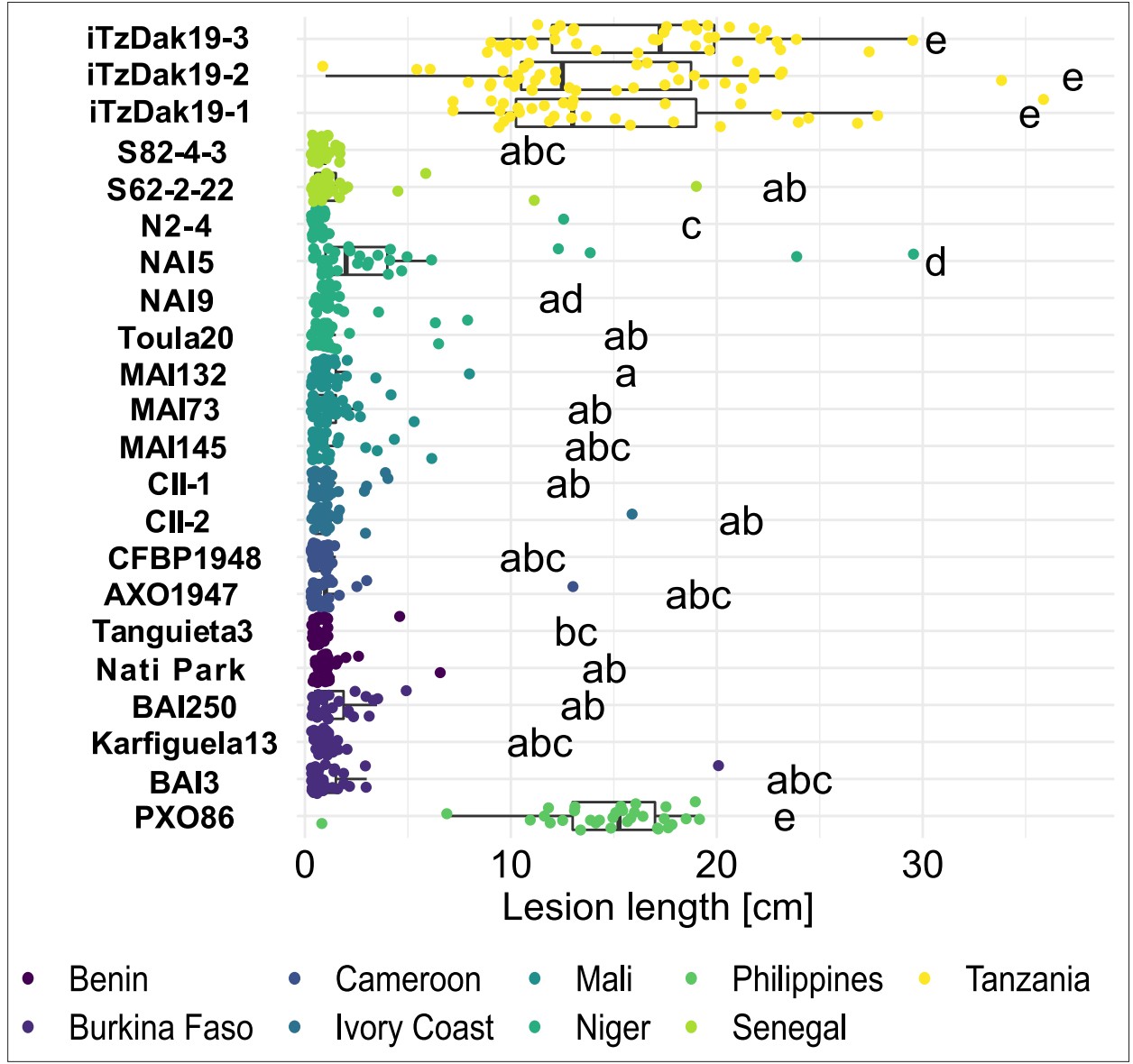

**Figure 2.** Resistance spectrum of wild-type Komboka rice against African *Xoo* strains. Leaf-clip inoculation of wild-type Komboka rice plants with a panel of 21 *Xoo* strains originating from 8 African countries along with Asian reference strain PXO86 from the Philippines. Lesion length (in cm) was measured 14 days after *Xoo* inoculation. Data from three independent experiments are represented. Letters in the boxplot represent significant differences. As a representative of the isolates from Lukenge iTzLuk21-3 was tested for virulence on Komboka and shown to be fully susceptible (*Figure 2—figure supplement 1*).

The online version of this article includes the following figure supplement(s) for figure 2:

**Figure supplement 1.** Comparison of the susceptibility of Komboka to Tanzanian strains iTzDak19-1and iTzLuk21-3.

**Figure supplement 2.** Virulence of the African *Xoo* diversity panel on the susceptible rice line Azucena.

**Figure supplement 3.** Presence of bacterial blight *R* genes in select rice varieties.

*Supplementary file 1a*). To evaluate the effectiveness of the resistance genes *Xa1*, *Xa3*, *Xa4*, *xa5*, *Xa7*, and *Xa23*, known to be efficient against African *Xoo* (*Gonzalez et al., 2007*; *Tekete et al., 2020*), six Near Isogenic Lines (NILs) of rice each harboring one of these single BB resistance gene, were inoculated with 18 endemic African strains and three strains isolated from Dakawa in 2019 (iTzDak19-1, iTzDak19-2 and iTzDak19-3; strain naming: Dak for Dakawa, Tz for Tanzania, and the novel strains collected in 2019 and 2021 obtained an 'i' for introduced) (*Table 1*; *Figure 2*). NILs harboring *Xa1*, *Xa23*, *xa5* or *Xa4* were resistant to most endemic African strains, but surprisingly, iTzDak19-1, iTzDak19-2, and iTzDak19-3 were virulent on all NILs tested, except IRBB23. These unusual iTz strains were therefore characterized in more detail (see below). The strains from Lukenge were isolated much later and could so far not be included in the race-typing analysis, but were also characterized in more detail as decribed below.

## Resistance of Komboka to endemic African *Xoo* but not to new Tanzanian strains

Komboka, an emerging elite variety released in several East African countries (Tanzania, Kenya, Uganda and Burundi) has been described as moderately resistant to *Xoo*, yet the nature of its resistance has not been elucidated. Using our African *Xoo* diversity panel, we found that most African strains were avirulent on Komboka, except for representatives of newly isolated strains from Dakawa (2019) and Lukenge (2021); (*Figure 2*, *Figure 2—figure supplement 1*; *Table 1*). By comparison, all strains were highly virulent on the susceptible variety Azucena (*Figure 2—figure supplement 2*; *Table 1*). Since *Xa1*, *Xa4*, *xa5*, and *Xa23* confer resistance against *Xoo* (*Table 1*), we investigated *R* gene presence in the Komboka genome. Mining of the IRRI QTL database (https://rbi.irri.org/resources-and-tools/qtl-profiles) revealed that Komboka contains genetic markers linked to *Xa4* but not *xa5* or *Xa23*, while there was no information on *Xa1* in the database (*Figure 2—figure supplement 3A*). We confirmed the presence of *Xa4* in Komboka by tracing an *Xa4*-associated marker and by sequencing (*Figure 2—figure supplement 3B,C*; *Supplementary file 1c*). Consistent with the broad-spectrum resistance of *Xa1* against endemic African *Xoo* strains, we found that Komboka carries a dominant *Xa1* allele that is highly similar to *Xa45*(t) (*Ji et al., 2020*; *Figure 2—figure supplement 3D, E*). The combination of *Xa4* and *Xa45(t)* in Komboka liekly can explain the observed resistance to most African *Xoo* strains. However, the *R* genes present in Komboka do not protect against the iTz strains recently isolated from the outbreak in Tanzania.

## Tanzanian *Xoo* strains cluster with Asian *Xoo* via whole genome SNP-based phylogeny

To identify the mechanisms underlying the virulence of the Tanzanian strains, three strains collected from Dakawa in 2019, and five from Lukenge in 2021, were subjected to whole-genome sequencing and SNP-based phylogenetic analyses (*Supplementary file 1a*, *Supplementary file 2*). Notably, all eight strains clustered with Asian rather than African *Xoo* isolates (*Figure 3A*). By contrast, older Tanzanian strains TzDak11-1, TzDak11-2 and TzDak18-1, which had been collected in Dakawa before 2019, grouped with the endemic African lineage, consistent with previous reports (*Oliva et al., 2019*). Based on the analysis of the core genome, the eight recently isolated strains carried only 1–4 core genome SNPs (*Figure 3—figure supplement 1*), intimating that they derive from a single introduction event. The phylogenetic analysis indicates that the new strains (named iTz; 'i' for *introduced*; Tz for Tanzania); are most closely related to strains from Yunnan province, China. Additional analyses will be necessary to identify the exact source of inoculum given the relatively small evolutionary distance between the strains from Yunnan and the iTz strains (*Figure 3B*). The new strains all carry *iTALe* genes similar to *tal3a* of PXO99[A] (*Figure 3C*). *iTALe* genes encode truncated TALes that suppress *Xa1* resistance, but had so far only been reported in Asian *Xoo* isolates. In addition, the strains contain a TALe (named PthXo1B) highly similar to the Asian PthXo1, with the addition of two RVDs at the N-terminus of the repeat array (*Figure 3D*; *Supplementary file 1d*). EBE prediction indicated that PthXo1B can bind to the *SWEET11a* promoter at a site overlapping with the EBE that is targeted by PthXo1 (*Figure 3D*). Consistent with a SWEET11a-based susceptibility, *ossweet11a knock-out* mutants from the diagnostic SWEET[R] kit were resistant to representative strains from the outbreak in Dakawa and Lukenge, iTzDak19-1 and iTzLuk21-3, whereas *ossweet13* or *ossweet14 knock-out* mutants remained susceptible (*Figure 4A*). Moreover, *SWEET11a* mRNA levels were elevated in Kitaake leaves after

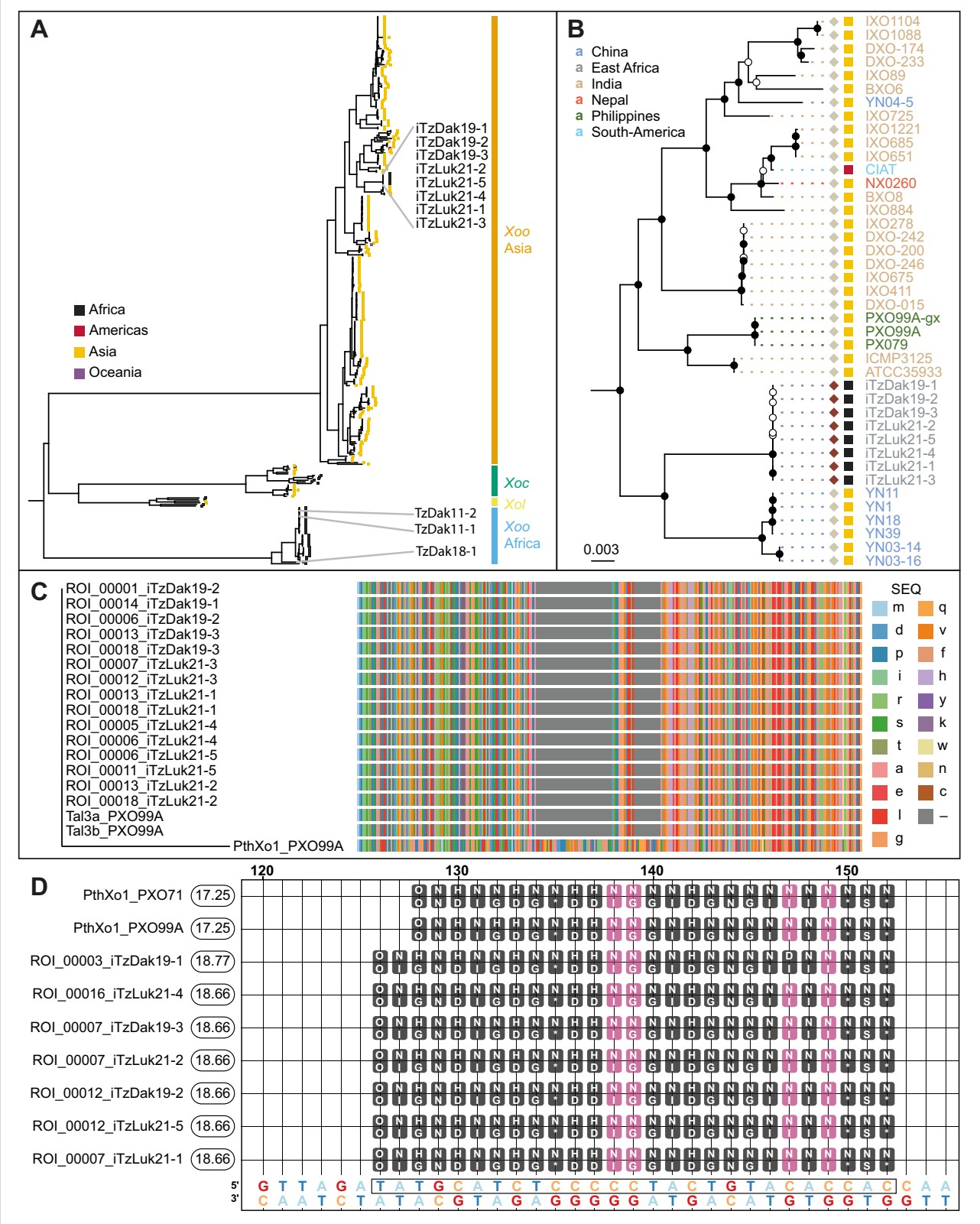

**Figure 3.** Analysis of the genomes of Tanzanian *Xoo* isolates. (**A**) Core genome *Xanthomonas oryzae* phylogenetic tree. Only the names of Tanzanian isolates are indicated. Abbreviations: *Xoo*, *X. oryzae* pv. *oryzae*; *Xoc*, *X. oryzae* pv. *oryzicola*; *Xol*, *X. oryzae* pv. *leersiae*. (**B**) Close-up view of the branches of the tree in A, including the newly isolated Tanzanian strains and neighboring clades. Black-filled nodes have a bootstrap support value equal to or above 80%. The scale bar reflects branch length in mean number of nucleotide substitutions per site. Colored squares reflect the continent of origin (as

*Figure 3 continued on next page*

*Figure 3 continued*

in A). Text color refers to the subregion of origin. (**C**) Multiple alignment of the N- terminal domain sequences of PXO99[A] iTALEs and the putative iTALEs from the eight newly isolated Tanzanian iTz strains. The -terminal domain of PthXo1 from strainPXO99[A] was used as canonical TALe. TALe references; lowercase letters represent amino acids. Gaps are colored gray. (**D**) Talvez EBE predictions on the *SWEET11a* promoter. Repeat Variable Di-residue (RVD) sequences (rounded boxes) are aligned along their predicted matching nucleotide along the promoter sequence (the lowest row). Black-filled RVDs match their target nucleotide in the Talvez RVD-nucleotide association matrix with the best possible score for this RVD. Those in violet match with an intermediate score. Values in the rounded boxes near the TALe names correspond to Talvez prediction scores. Sequences are provided in *Figure 3— source data 1*.

The online version of this article includes the following source data and figure supplement(s) for figure 3:

**Source data 1.** Compressed archive with Fasta Sequence files of *Xoo* strains CIX4462, CIX4506 and CIX4462.

**Figure supplement 1.** Pairwise counts of SNPs in the core genome of the newly isolated *X. oryzae* pv. *oryzae* strains from Tanzania and the closest neighboring clade.

inoculation with iTzDak19-1 and iTzLuk21-3 (*Figure 4B*). Together, our data show that the iTz strains collected in 2019–2021 in the Morogoro region are >99.99% identical to each other, and phylogenetically related to Asian *Xoo* strains. The strains contain both *iTALe* and *pthXo1* homologs unique to Asian *Xoo* isolates, providing insights into the virulence mechanism and providing a basis for the development of approaches that can protect African rice varieties, and in particular Komboka, against the novel iTz strains found in the outbreak (*Oliva et al., 2019*).

## Editing *SWEET* promoter sequences in Komboka to obtain resistance to Asian and African strains

As a prerequisite for editing *SWEET* promoters in Komboka, the promoter regions of *SWEET11a*, *13* and *14* were sequenced. The promoters contain EBEs for PthXo1 and PthXo1A, PthXo2A, PthXo3, AvrXa7, TalC, and TalF, respectively (*Figure 5*; *Figure 5—figure supplement 1*, *Supplementary file 1e*). To protect Komboka against endemic strains from Africa as well as introduced Asian strains, all six known EBEs in the promoters of *SWEET11a*, *13* and *14* were edited using a hybrid Cas9/ Cpf1 editing system. Due to blunt cleavage, Cas9 preferentially produces SNPs, while Cpf1 (Cas12) produces staggered cuts, creating predominantly small deletions. We hypothesized that small deletions may produce more robust resistance because the sequences would differ more from the target EBE. Moreover, the probability of obtaining combinations of optimal mutations in all EBEs is expected to be higher compared to Cas9 approaches (*Oliva et al., 2019*). Due to the PAM requirement, it was not possible to design sgRNAs for Cpf1 at all EBEs; therefore, Cpf1 was combined with Cas9. Two CRISPR/Cpf1 gRNAs (cXo1 and cXo2) were designed to target PthXo1 /PthXo1 A and PthXo2A EBEs in *SWEET11a* and *SWEET13*, respectively (*Figure 5—figure supplements 1 and 2*; *Supplementary file 1f*). Since the EBEs for AvrXa7, PthXo3 and TalF in *SWEET14* are overlapping, one CRISPR-Cpf1 gRNA (cTalF) was designed to target the overlapping region of all three EBEs. Because no TTTV PAM sequence was available near the EBE for TalC for designing a gRNA required by Cpf1, a Cas9 gRNA (gTalC) was designed to target the TalC EBE. From a first round of transformation, six representative T2 lines were tested for resistance using six representative *Xoo* strains: ME2 (PXO99[A] mutant deficient in PthXo1), PXO99[A] (PthXo1), PXO61 (PthXo2B, PthXo3), PXO86 (AvrXa7), MAI1 (TalC, TalF), and BAI3 (TalC) (*Figure 5A*, *Figure 5—figure supplement 3*, *Supplementary file 1g-k*). All six lines from this first round of transformation were resistant to the tested strains. One line (1.5_19) was fully resistant to all strains, while five lines were fully resistant to five strains, but only moderately resistant to PXO86. This difference in resistance is most likely explained by the presence of a rather small 4 bp deletion in the EBE for AvrXa7, while line 1.5_19 carried the same 4 bp deletion plus an additional base pair substitution (G/T) (*Figure 5B*). Since the clipping assays us very high bacterial titers, it is generally assumed that moderate resistance will be sufficient for effective resistance in field conditions (*Adhikari et al., 1995*; *Fred et al., 2016*). However, TALes were reported to have less specific nucleotide binding at the 3'-end, hence the G/T substitution in line 1.5_19 could potentially be overcome by adaptation of AvrXa7 (*Richter et al., 2014*). To obtain more robust resistance to all known *Xoo* strain, a second round of transformation was carried out. Two T2 lines (14_19 and 14_65), which contained deletions in all EBEs, including 11- and 5 bp deletions in the AvrXa7 EBE, respectively, were resistant to all strains tested, including PXO86 (*Figure 6*). Lines 1.5_19 and 1.2_40 were also resistant to iTzDak19-1 and iTzLuk21-3, consistent with the 12- and 9 bp deletions in the predicted PthXo1

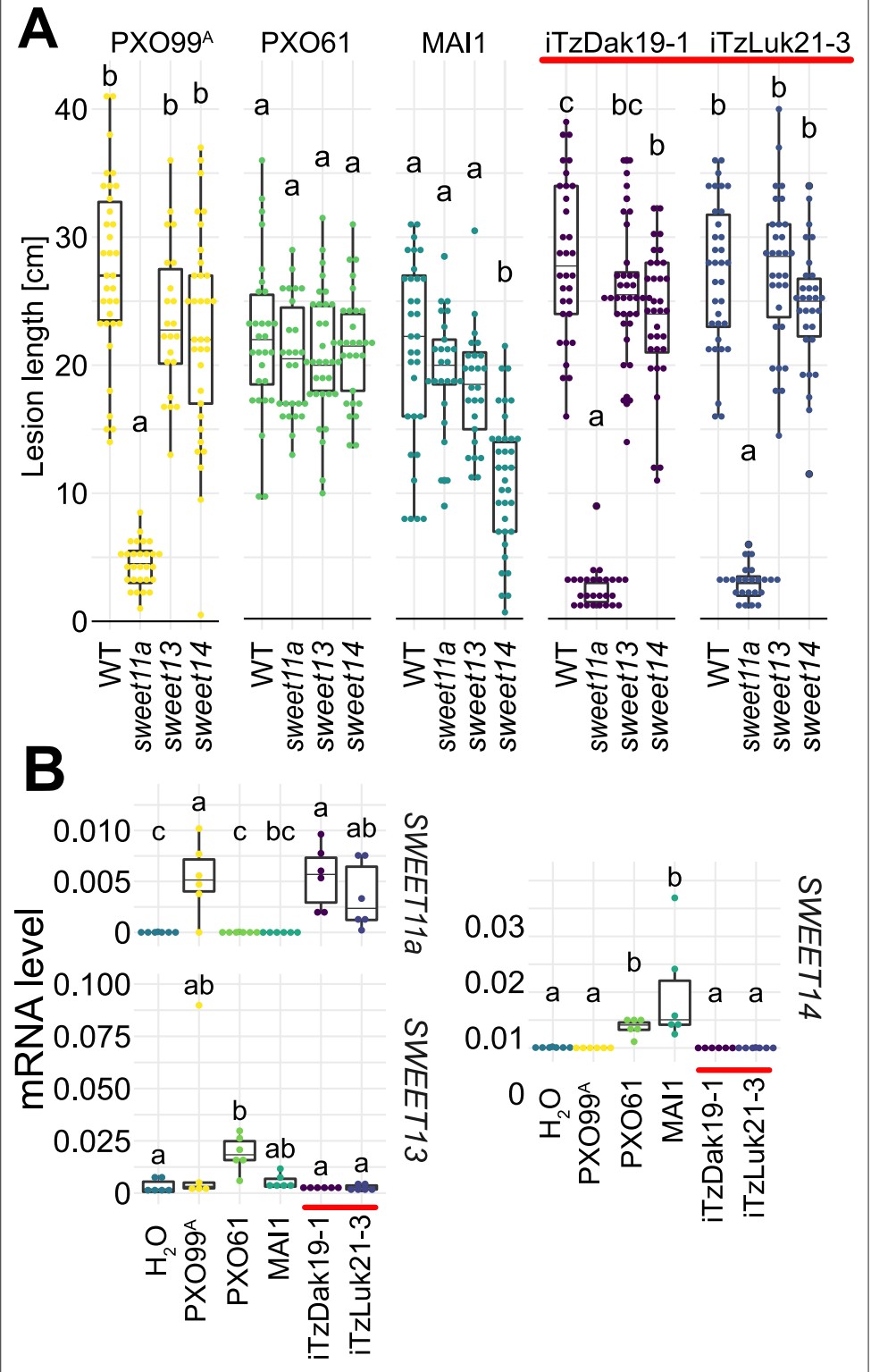

**Figure 4.** Virulence of the new Tanzanian *Xoo* strains depends on the induction of *SWEET11a*. (**A**) Lesion lengths were measured 14 days after leaf-clipping inoculation of Kitaake individual *sweet* knock out lines (oss*weet11a*, oss*weet13* and oss*weet14*) in the cultivar Kitaake (***Eom et al., 2019***) with PXO99[A] (PthXo1), PXO61 (PthXo2B/ PthXo3), MAI1 (TalC/TalF), and Tanzanian strains (iTzDak19-1 and iTzLuk21-3) from the recent outbreaks (highlighted by a red bar). Results from two independent experiments are represented. (**B**) Relative mRNA levels (2[-ΔCt]) of *SWEET11a*, *SWEET13* and *SWEET14* in wild-type Kitaake upon infection by PXO99[A] (PthXo1), BAI3 (TalC),

*Figure 4 continued on next page*

*Figure 4 continued*
and Tanzanian *pthXo1B* dependent *Xoo* strains. Samples were collected 48 hr post infiltration. Data from three independent experiments were pooled and are represented together. Ct values were normalized to the rice *EF1α* elongation factor (ΔCt).

EBE (*Figure 7*). As one may have predicted, induction of *SWEET11a* by iTzDak19-1 and iTzLuk21-3 was abolished in lines 1.2_40 and 1.5_19 (*Figure 7—figure supplement 1*). Taken together, Komboka lines with full resistance to representative Asian and African *Xoo* strains were obtained; notably with resistance to two representative strains from the emerging iTz population from Dakawa and Lukenge.

## Discussion

Here, we report a rapidly spreading outbreak of BB in East Africa that was first identified in 2019 in the Morogoro region in Tanzania. Through genetic and genomic characterization, the outbreak can be ascribed to a recent colonization event by *Xoo* strains in the Morogoro region that are most similar to Asian strains, that is, from Yunnan province in China. Notably, strains that originated from Asia and Africa before 2019 form distinct phylogenetical groups (*Lang et al., 2019*; *Tran et al., 2018*). Before 2019, BB incidence was relatively low, and only three strains were isolated between 2011 and 2018 (*Oliva et al., 2019*). As shown previously, these three endemic Tanzanian strains cluster with other African *Xoo* strains, mostly from West Africa, based on SNP-based core genome phylogeny analyses, and all African strains make use of TalC as the major virulence TALe, some additionally use TalF to trigger SWEET14 induction (*Oliva et al., 2019*). At present, we cannot trace the exact history of introduction of the new Asian-type strains, which may have occurred through either major storms, animal migration or transfer of contaminated rice seeds. *Xoo* can be transmitted by tropical storms. Evidence for transmission within Asian countries has been provided, for example, for PXO99[A], which may have been introduced rather recently from Nepal or India to the Philippines. By contrast, rather effective continental isolation had been observed. One may hypothesize that if the new Tanzanian strains had been transmitted by storms from Asia, the strains might be more closely related to strains from India or Indonesia based on the geography (*Oliva et al., 2019*; *Quibod et al., 2016*; *Tran et al., 2018*). Due to the distinct evolutionary trace and the resulting absence of coevolution, it is likely that many African rice varieties do not contain suitable *R*-genes that protect against Asian strains. Here we identified the dominant *R*-gene *Xa23* as being effective against the iTz strains. One may thus consider to introgress into local rice varieties in East-Africa. Based on the sequence information for the iTz strains and resistance profiling of Chinese strains, a combination of *Xa21* and *xa13* might also be able to protect against the iTz strains.

Hallmark features of the new strains identified in Tanzania are the presence of iTALes and the PthXo1 homolog PthXo1B; both are absent from endemic African strains. African rice lines carrying *Xa1* may thus be resistant to endemic African strains, but will be susceptible to Asian strains, including those recently introduced to Tanzania. Surveys over several years, including preliminary data from 2022, indicate that the outbreak is gaining momentum regarding severity and spread, thus potentially becoming a threat to East Africa, and over the course of time, possibly to all of Africa. This study, which so far is based on the full analysis of only a few strains from two locations in Morogoro, shows that eight strains are genetically almost identical, intimating a single introduction event in this region. However, further analysis of a greater number of strains will be required to validate origin and relationships, as well as further evolution.

We had previously described that ectopic induction of at least one clade III *SWEET* by a TALe is essential for virulence of all *Xoo* strains characterized to date (*Chen et al., 2012*; *Eom et al., 2019*; *Oliva et al., 2019*; *Streubel et al., 2013*). A major difference between Asian and African strains is the distinct set of TALes that target SWEETs, in which endemic African strains specifically target only distinct EBEs for TalC and TalF in SWEET14 (*Oliva et al., 2019*). It is thus likely, due to the absence of coevolution, that many African rice varieties contain EBEs in *SWEET11a, 13 and 14* promoters that can be targeted by Asian TALes. In Asia, single *R* genes and even combinations of several *R* genes are insufficient for robust resistance, because new strains evolved that break individual *R* genes or combinations thereof (*Arra et al., 2017*). Therefore, breeders test current strains against a panel of rice NILs that carry individual, or combinations of different *R* genes (http://www.knowledgebank.irri.

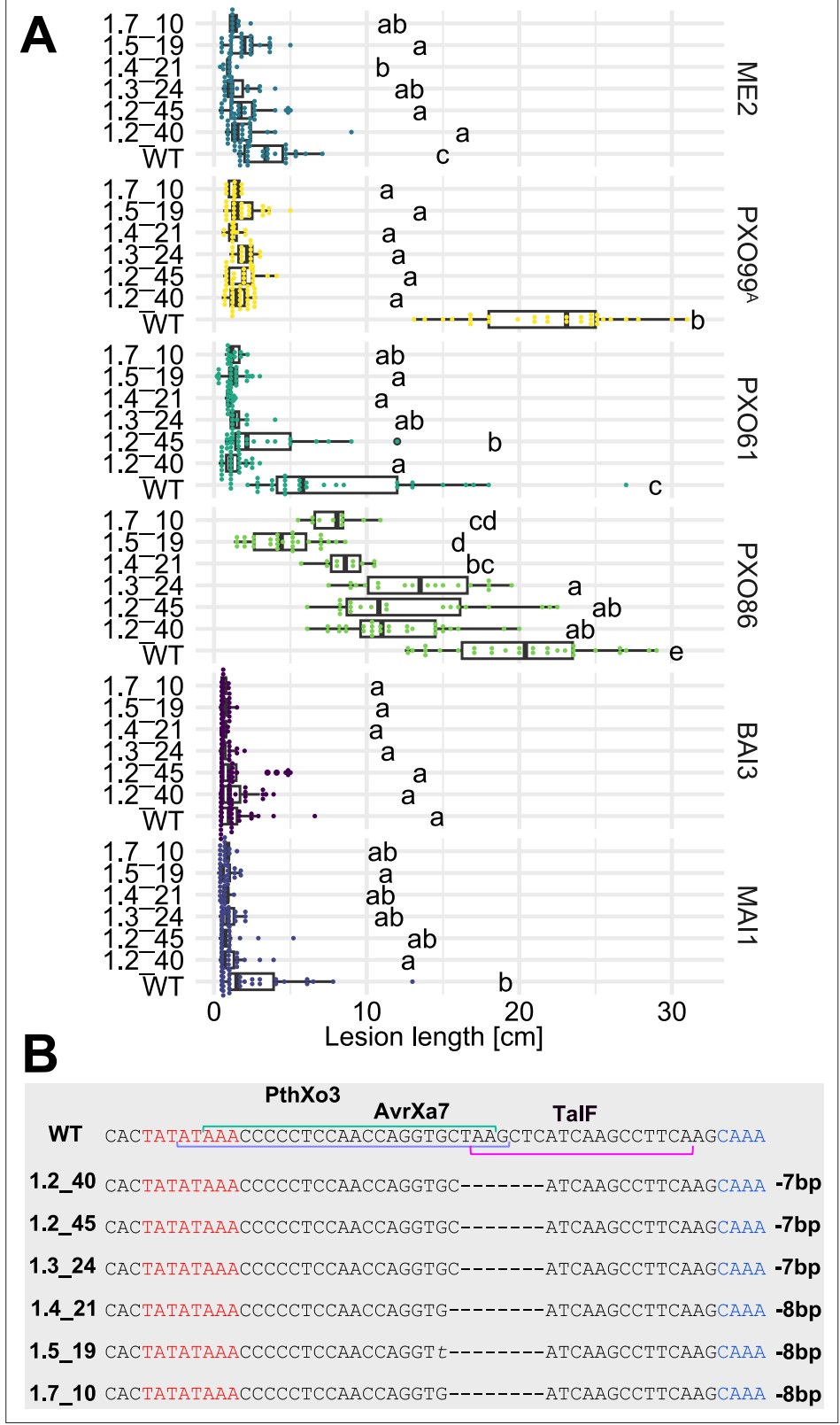

**Figure 5.** Resistance of EBE-edited Komboka lines against six representative *Xoo* strains. (**A**) Reactions of WT and six edited Komboka lines to infection by *Xoo* strains (PXO99[A], PXO61, PXO86, BAI3, and MAI1) harboring PthXo1, PthXo2 PthXo3, AvrXa7, TalC and/or TalF. ME2 is a PXO99[A] mutant strain with mutant deficient in PthXo1 and served as a negative control. The Komboka lines were carried edits in all six EBE sites validated by DNA

*Figure 5 continued on next page*

*Figure 5 continued*

sequencing.(**B**). Details of the EBEs targeted by PthXo3, AvrXa7 and TalF, and respective mutations in the six edited lines.

The online version of this article includes the following figure supplement(s) for figure 5:

**Figure supplement 1.** Comparison of EBE sequences for the *SWEET11a*, *13* and *14* promoters between Komboka and Kitaake.

**Figure supplement 2.** Hybrid CRISPR-Cas9/Cpf1 system.

**Figure supplement 3.** Genotypes of T0 lines with biallelic mutations at six targeted EBEs.

---

org/ricebreedingcourse/Breeding_for_disease_resistance_Blight.htm)(*Padmaja et al., 2017*). Subsequently information on the optimal set of *R* gene combinations is used to stack *R* genes to obtain the broadest possible spectrum of resistance against local populations of the pathogen (*Arra et al., 2018*). Researchers and breeders use clipping assays with high bacterial titers to evaluate resistance of rice lines to BB (*Kauffman et al., 1973*). The assay, even when performed in greenhouses, is considered to be highly predictive of field performance of resistant varieties (*Adhikari et al., 1995*; *Fred et al., 2016*; *Padmaja et al., 2017*). Due to the full dependence of *Xoo* virulence on the activation of particular *SWEET* genes, a complete block of induction of these *SWEET* genes by introducing mutations into the EBE is deemed the most effective way to protect rice varieties against the currently known spectrum of Asian and African *Xoo* strains.

Due to the long timeline between identification of an elite variety, that is, the need to establish suitable transformation protocols, the actual genome editing, elimination of transgenes, transfer to countries that instated regulations for SDN-1 based genome editing, field testing and registration, target varieties have to be chosen many years before deployment. Komboka was chosen since it was predicted to become popular in Kenya, a country that established guidelines that allow the use of edited lines, provided they do not contain transgenes (*Buchholzer and Frommer, 2023*). While a particular rice variety may contain one or several *R* genes, they can be overcome by existing mechanisms, thus introduction of the combined SWEET EBE mutations will likely provide a much broader and more robust resistance to BB. Analysis of Komboka, a high yielding semi-aromatic rice line released in Kenya and Tanzania, showed resistance to endemic African strains due to *Xa1* and *Xa4*. However, both *R*-genes had previously been broken in Asia. Notably, Komboka was found to be susceptible to Asian strains like PXO99[A], which carries both iTALes and PthXo1 that suppress *Xa1* and induce *SWEET11a*, respectively. Indeed, we found Komboka samples with BB symptoms in one district in Tanzania (Mikindani, Mtwara, ~400 km from Dakawa) in 2022. Characterization of the collected leaves is ongoing. *Xa1*-mediated resistance has been broken by 95% of the Asian *Xoo* strains (*Ji et al., 2020*), and *Xa4*-mediated resistance was overcome by the new East African *Xoo* strains as well as many Asian strains reported previously (*Quibod et al., 2016*). Thus, *Xa1* and *Xa4* lack durability due to shifts in the *Xoo* population or the emergence or introduction of new strains. Thus, new lines with broad-spectrum resistance against both Asian and African strains, including the strains found in Tanzania, is required (*Quibod et al., 2020*; *Vera Cruz et al., 2000*).

We hypothesized that editing all known EBEs for both African and Asian SWEET-targeting TALes could be used to engineer full and robust resistance in an emerging elite variety for Africa. Since this strategy requires simultaneous editing of six EBEs, optimally causing small deletions in each EBE to enhance robustness, we developed a hybrid CRISPR-Cas9/Cpf1 system that targets all six EBEs in Komboka. Both CRISPR-Cas9 and CRISPR-Cpf1 systems have been widely used in genome editing. SpCas9 is the first Cas version to offer high editing efficiency across a wide range of plant species. LbCpf1 or LbCas12 is another endonuclease which has a lower molecular weight compared to SpCas9 and requires shorter CRISPR RNA (crRNA)(*Liu et al., 2017*). SpCas9 and LbCpf1 both generate double-stranded breaks, but Cas9 introduces blunted cuts 3 base pairs upstream of the PAM, while Cpf1 introduces 5 bp staggered cuts downstream of the PAM (*Zetsche et al., 2015*). Hence, Cpf1 often produces larger mutations compared to Cas9, which mainly introduces single base pair insertions or deletions (indels). Depending on the purpose of genome editing, larger or smaller alterations may be preferred. To generate knock-out mutants, single base pair indels in the first exon that result in frameshifts or premature stop codons are sufficient. To prevent the binding of TALes to their respective EBEs on *SWEET* promoters, single base pair changes are not always effective due to

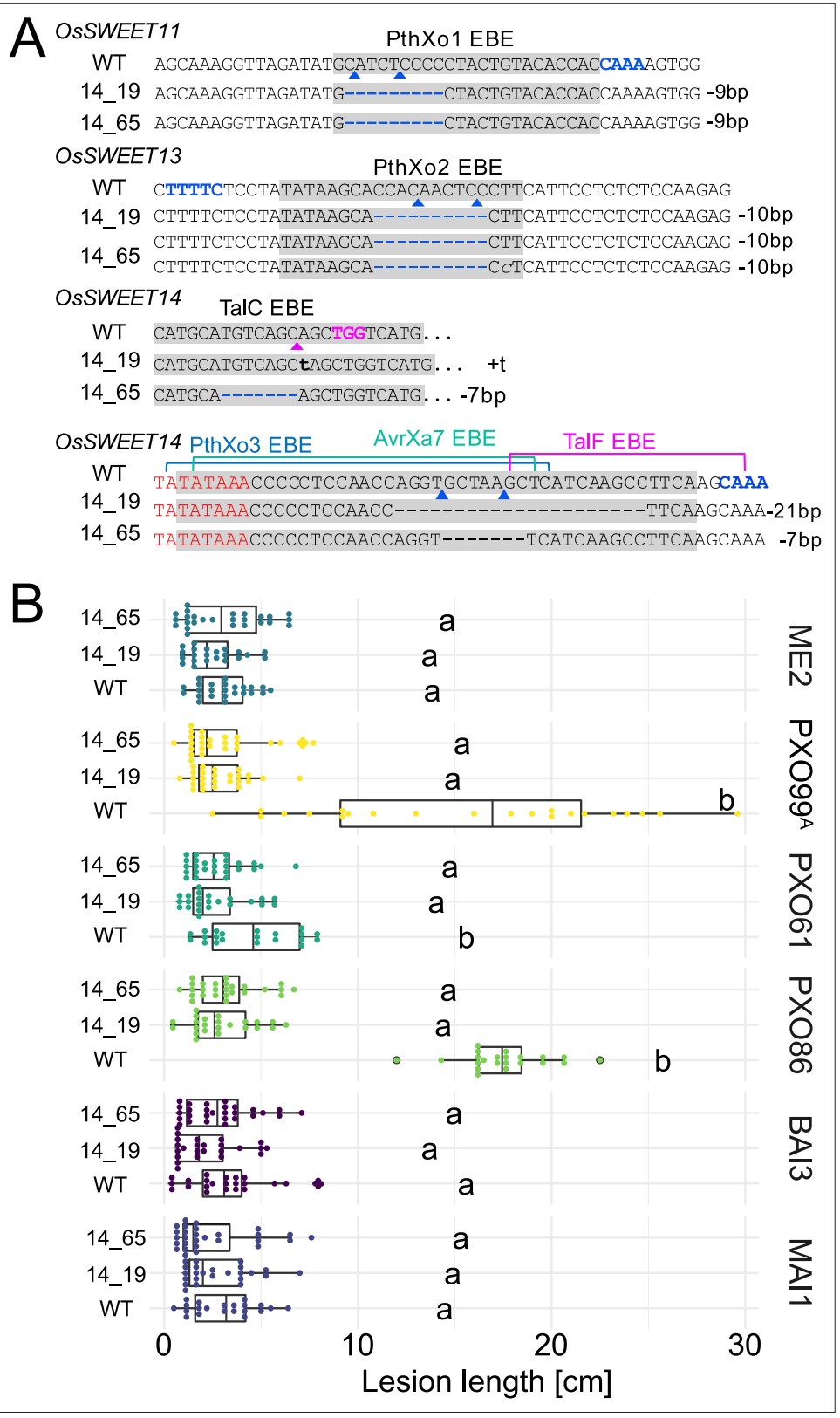

**Figure 6.** Genotypes and phenotypes of two EBE-edited lines generated in a second transformation experiment. (**A**) Mutations at the EBEs for PthXo1, PthXo2, TalC, TalF, PthXo3, and AvrXa7 in two Komboka edited lines, 14_19 and 14_65. (**B**). Reactions of wild-type Komboka and the two edited lines to the infections by PXO99[A] (PthXo1), PXO61 (PthXo2B, PthXo3), PXO86 (AvrXa7), BAI3 (TalC) and MAI1 (TalC, TalF). PXO99[A]ME2 (ME2) is a PXO99[A] mutant carrying a transposon insertion in *pthXo1* and served as a negative control.

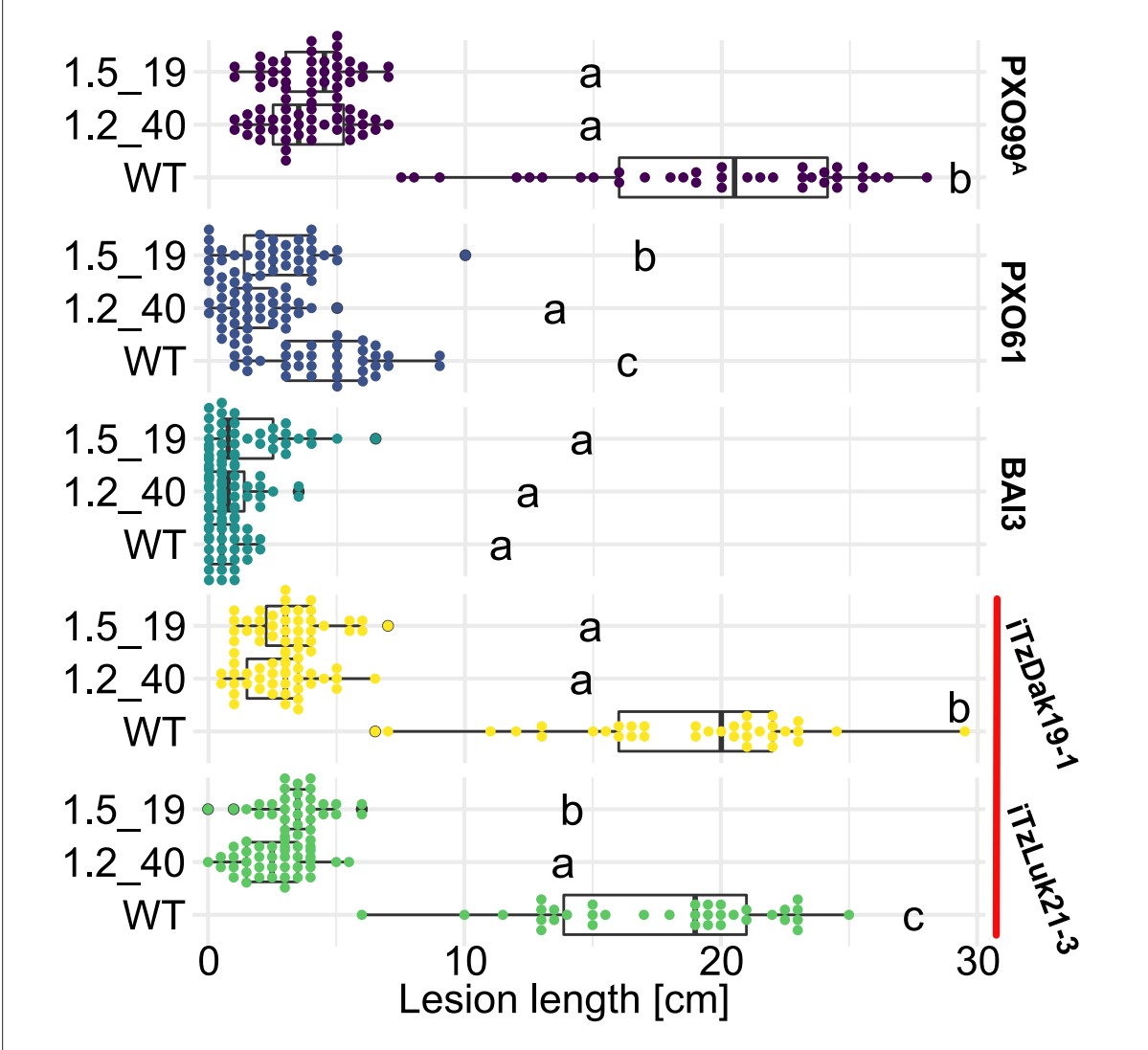

**Figure 7.** Komboka *SWEET* promoter edited lines are fully resistant against Tanzanian strains of *Xoo*. Lesion lengths measured 14 days after leaf-clip inoculation of wild-type Komboka and the two multi-edited lines 1.2_40 and 1.5_19 with *Xoo* strains PXO99[A] (PthXo1), BAI3 (TalC), PXO61 (PthXo2B, PthXo3), and the newly isolated Tanzanian strains iTzLuk21-3 and iTzDak19-1 (both carrying PthXo1 homologs and iTALes, red bar). Data from four independent experiments are represented.

The online version of this article includes the following figure supplement(s) for figure 7:

**Figure supplement 1.** qRT-PCR analysis of *SWEET11a* mRNA accumulation of edited Komboka lines after infection with Tanzanian *Xoo* strains.

the degenerate code and the ability to loop out central RVD repeats of the TALes (*Oliva et al., 2019*). In addition, *Xoo* evolves rapidly and, therefore, single base pair changes could be overcome quickly, as demonstrated for an *xa25*-like resistance gene (*Xu et al., 2019*). Hence, to prevent rapid break of the resistance by simple modifications of the TAL effectors, a larger mutation would be preferred. Since it is still unknown how the native *SWEET* expression is affected by promoter mutations, the number of base pair indels needs to be evaluated carefully. In addition, prevention of TALe binding to EBEs depends not only on the number of deleted or inserted base pairs but also on the position of the introduced mutations within the EBE. Especially, the RVDs at the 3´-end of TALes, such as N* and NS, have less specificity in nucleotide binding, therefore, mismatches at the 3´-end of an EBE can be tolerated by the RVDs (*Richter et al., 2014*). Here, the 1 bp deletion at the 5´-end of the TalC binding site is sufficient for resistance, while a 4 bp deletion at the 3´-end of the AvrXa7 binding site did not

prevent binding of AvrXa7 to its EBE, indicating that mutations at the 5′-end of the TALe binding site are more effective than mutations at the 3′-end in preventing the binding of TALes to EBEs.

Cas9 cleaves target DNA adjacent to a G-rich PAM, while Cpf1 requires a T-rich PAM sequence (*Zetsche et al., 2015*; *Zhang et al., 2014*). Therefore, combining Cas9 and Cpf1 into a single system enables simultaneous editing in both G- and T-rich regions. Here, EBE regions used for gRNA design were comparatively short, just 23–29 bp, hence, there were a limited number of suitable gRNA design options (considering also potential unwanted off-target effects). The BB-resistant IR64 and Ciherang-Sub1 lines developed previously for Asia were generated using four gRNAs for *SWEET11a*, *13* and *14* promoters, but did not target the TalF EBE in *SWEET14* which is targeted by African strains specifically (*Eom et al., 2019*; *Oliva et al., 2019*). The EBE in the respective promoters for TalF lacked a suitable SpCas9 NGG-PAM recognition sequence. By combining the two enzymes Cpf1 and Cas9 into a hybrid CRISPR-Cas9/Cpf1 system, it became possible to design four gRNAs (three gRNAs for Cpf1, including one for TalF; and one for Cas9 to target TalC, which lacked a suitable PAM sequence for Cpf1) that target all six known EBEs relevant for BB susceptibility. The combination of CRISPR-Cas9 and Cpf1 gRNAs in a single system increases the flexibility of choosing suitable gRNAs depending on the available PAM recognition sequences and off-target frequency. An efficient transformation protocol previously established for Komboka was used here to edit all known EBEs in three *SWEET* promoters (*Luu et al., 2020*). For four target sites (cXo1, cXo2c, gTalC, cTalF), we generated 15, 27, 16, and 21 variants, respectively using four gRNAs, demonstrating the successful application of the hybrid Cas9/Cpf1 in multiplex genome editing of the rice variety Komboka. Each variant represents a distinct *R* gene. Edited Komboka lines were shown to be resistant against a set of representative Asian and African *Xoo* strains, which harbor all known *SWEET*-inducing TALes.

It has been suggested that TALes can evolve rapidly to allow adaption to host resistance due to changes in the respective EBEs. It is thus conceivable that the resistance developed here can be broken by the evolution of new TALe variants that recognize elements in the edited SWEET target promoters (*Perez-Quintero and Szurek, 2019*). Various observations may indicate however that the velocity at which such new TALe emerge is slower than expected: although all six Clade 3 *SWEET*s can theoretically serve as susceptibility factors as shown by using artificial artTALes, neither African nor Asian strains seem to target *SWEET11b*, *12*, or *15* yet (*Streubel et al., 2013*; *Wu et al., 2022*). Notably, *xa13*, carrying a large promoter insertion in *SWEET11a*, is still used widely; thus we hypothesize that no new TAL effectors that target SWEET11a have emerged and are able to break the resistance. Notwithstanding, even if such variants emerge, new promoter variants can be developed by editing at time scales not too different from that of the spread of the new strains.

Crosses and analyses for determining the reliable elimination of transgenes from the edited Komboka lines described here is in progress. In parallel, monitoring the evolution of *Xoo* in Africa is of critical importance for early detection of Asian strains, hybrid African-Asian strains or newly emerging strains. The diagnostic SWEET^R Kit 2.0, upgraded with the newly identified *SWEET11b* susceptibility gene and containing all six translational SWEET-reporter and knockout lines, can effectively be used to detect the targeted EBEs, while TALome analysis can be performed to detect the presence of *SWEET*-inducing TALes in the current African *Xoo* population (*Eom et al., 2019*; *Wu et al., 2022*). Kenya evaluates applications for the import of transgene-free lines on a case-by-case basis, thus it will be necessary to prepare suitable documentation for import. In parallel, it is planned to use the hybrid Cas9/Cpf1 strategy to expand the resistant germplasm to other rice varieties grown in African countries to be able to meet consumer-farmer preferences. This project aims to ultimately provide the material to small-scale producers. Governmental action is recommended to reduce the risk of introduction of Asian pathogens to Africa and vice versa. In parallel, improved management practices and training may also help farmers to reduce disease incidence (*Mew et al., 2018*).

## Methods
### Rice seeds and *Xoo* strains
Seeds of the rice variety Komboka (IR05N221, L17WS.06#24) were provided by the International Rice Research Institute (IRRI, The Philippines) under a Standard Material Transfer Agreement under the Multilateral System (SMTA-MLS). *Xoo* strains were obtained from the *Xoo* strain collection of at the French National Research Institute for Sustainable Development (IRD, Montpellier, France).

## Rice cultivation

Rice seed germination and plant cultivation at HHU were done as described (*Luu et al., 2020*). Briefly, rice seeds were sterilized and germinated onto ½ Murashige Skoog media (Duchefa, M0222.0050), supplemented with 1% sucrose (Sigma-Aldrich, S7903-250G). The seedlings were grown for 10 days in magenta boxes before transferring to soil. The plants were grown in greenhouses (8 hr day 30 °C / 16 hr night 25 °C, relative humidity (RH) 50–70%, supplemental LED (Valoya, BX100 NS1) at 400 µmol/m$^{-2}$s$^{-1}$). The plants were fertilized weekly from the 2nd week and biweekly from the 6th week after germination (ICL, Peters Excel, CalMag grower, 2152.02.15EB). At IRD, plants were sown in soil complemented with 3 g of standard NPK fertilizer (N: 19%; P: 5%; K: 8%) per liter and grown in green-houses (12 hr day 28 °C and 80% RH, 12 hr night at 25 °C and 60% RH at 200 µmol/m$^{-2}$s$^{-1}$).

## Disease resistance scoring by leaf-clipping inoculation

Resistance/susceptibility was assessed using leaf clipping assays (*Kauffman et al., 1973*). This assay introduces an effective entry for bacteria through a massive wound, providing access to the xylem vessels for an inoculum of Xoo in the logarithmic phase at extremely high titers of ~10$^8$. This assay is used by breeders and is, due to the extreme exposure of the plant, highly predictive for resistance in field conditions. Breeders and leading BB experts we contacted were not aware of cases in which resistance as determined by the clipping assay in greenhouse conditions did not translate to resistance in paddy field conditions. Notably, lines with moderate resistance lesion length of up to 5 cm in the clipping assay in greenhouses were fully resistant under field conditions. Multiple publications support the reliability and predictive efficacy of Kauffman clipping assays for field performance of resistance (*Adhikari et al., 1995*; *Fred et al., 2016*; *Kauffman et al., 1973*; *Padmaja et al., 2017*). Bacteria were grown on PSA media (Peptone Sucrose Agar, 1% peptone (Gibco, Bacto Peptone, 211677), 1% sucrose (Sigma-Aldrich, S7903-250G), 0.1% glutamate (Sigma-Aldrich, G8415-100G), 1.6% agar (Sigma-Aldrich, 05040–1 KG)) for 4 days at 28 °C. Single colonies were picked and patched onto PSA and then grown for 24 h before being washed with and diluted in sterile water at OD$_{600}$ 0.2 (at IRD) and 0.5 (at HHU). The youngest, fully extended leaves of 4- to 6-week-old rice plants were clipped 2–3 cm from the leaf tip with scissors that had been dipped in the inoculum or sterile water. Lesion lengths were measured 14 days after inoculation. For the race typing assay, lesion length measurements <5 cm were scored as resistant (R), 5–10 cm as moderately resistant (MR), 10–15 cm as moderately susceptible (MS), and >15 cm as susceptible (S).

## Sampling of symptomatic leaf material and isolation of *Xoo* strains

Small-scale rice fields cultivated under irrigated conditions in the villages of Dakawa and Lukenge (in the Morogoro region of Tanzania) were surveyed for BB between 2019 and 2021. Larger scale monitoring of the disease was achieved all over the country in 2022 and processing of these samples is in progress (*Figure 1*). In each field, diseased leaves of several individual plants were sampled and processed for bacterial isolation as reported previously (*Tekete et al., 2020*). Colony-multiplex PCR was used to validate that isolates were *Xanthomonas oryzae* pv. *oryzae* (except for 2022 data, for which analyses have been initiated).

## Disease scoring in Tanzania

Disease scale/diseased leaf area of BB was scored by breeders (previous knowledge of breeders from Tanzania, who were trained as per IRRI SES, 5th edition) using IRRI's Scoring System for field tests of Evaluation of BB Resistance (http://www.knowledgebank.irri.org/ricebreedingcourse/Breeding_for_disease_resistance_Blight.htm) (*Ardales et al., 1996*; *Vera et al., 1996*).

## Greenhouse tests

| Lesion length | description |
|---|---|
| 0–5 | R |
| >5–10 | MR |

*Continued on next page*

*Continued*

| Lesion length | description |
|---|---|
| >10–15 | MS |
| >15 | S |

## Field tests

| Scale | diseased leaf area (%) | description |
|---|---|---|
| 1 | 1–5 | R |
| 3 | 6–12 | MR |
| 5 | 13–25 | MS |
| 7 | 26–50 | S |
| 9 | >50 | S |

Note that there is a sampling bias due to the expansion of the evaluation sites in 2022. Notably, before 2019, BB had been observed but damage was not severe enough to be monitored systematically in Dakawa and Lukenge, nor in any other parts of Tanzania. Also note that the surveys were not carried out over longer periods in the three years and provide only estimates of severity at the time of observation. Severity will, among other factors, depend on the time of initial infection of the field, the bacterial titer, and the stage at which infection occurred. Moreover, the time of observation also affects the interpretation, that is, at late stages symptoms may not be recorded since the plants are senescent and symptoms cannot be recorded on senescent leaves.

## Tanzanian strain genome sequencing and analysis

DNA was extracted from pure bacterial cultures grown on rich media (1% peptone [Gibco, Bacto Peptone, 211677], 0.1% glutamate [Sigma-Aldrich, G8415-100G]) with QIAGEN Genomic-tip 100 /G (Qiagen, Hilden, Germany, 10243). Multiplex libraries were prepared with the rapid library preparation kit (SQK-RBK110-96, Oxford Nanopore Technologies, ONT) and sequenced with a MinION Mk1C device on R10.3 (F Oxford Nanopore Technologies, LO-MIN111) flow cells. ONT electric signals were base-called with a high accuracy model (dna_r10.3_450bps_hac.cfg) and demultiplexed with the ONT Guppy base calling software (v6.0.1+652ffd179). Illumina library construction and sequencing was performed by FASTERIS (Plan-les-Ouates, Switzerland) on an Illumina NextSeq sequencer with 350 bp inserts and 150 bp paired-end sequences. Long ONT read assembly was performed with the CulebrONT pipeline (v2.0.1) (*Orjuela et al., 2022*) and included FLYE (2.9-b1768) for primary assembly followed by RACON (v1.4.20) and MEDAKA (1.4.1) for assembly polishing. The ONT-only assemblies were subsequently polished with polypolish (v0.5.0) (*Wick and Holt, 2022*). Core genome SNP genotyping and tree reconstruction were conducted as previously described (*Doucouré et al., 2018*), except that the raxml command included '-N 500 m GTRGAMMAIX'. This substitution model was adopted based on the output of the 'modelTest' function of the R package phangorn for model selection (*Schliep, 2011*). The X11-5A assembly (GB Acc. GCF_000212755.2) was used as an out-group, and was not displayed in *Figure 3*. The genomes included in the species-wide *Xanthomonas oryzae* phylogenetic tree are listed together with NCBI accessions and genome metadata in *Supplementary file 2* and *Figure 3—source data 1*. Predictions for *TALe* genes from the genomes of the Tanzanian strains used AnnoTALE with default parameters (*Grau et al., 2016*). Predictions for high-scoring EBEs of the Tanzanian TALes in the Nipponbare *SWEET11a* promoter used Talvez with default parameters (*Pérez-Quintero et al., 2013*). Talvez target predictions on the Nipponbare SWEET11a promoter included putative TALes from Tanzanian strains (iTzDak19-3, iTzLuk21-1, iTzLuk21-2, and the endemic strains that predate the current outbreak: TzDak11-1, TzDak11-2, and TzDak18-1), Asian strains (PXO99[A], PXO86, PXO83, PXO71, PXO61) and an African strain (MAI1), which were included as references. Only predictions with a score >9.05 are displayed in *Figure 3*.

## qRT-PCR analyses of *SWEET11a* mRNA levels

Leaves of 3-week-old rice plants were infiltrated with a bacterial suspension at an $OD_{600}$ of 0.5 or water using a needleless syringe. One leaf per plant was infiltrated, and three plants per treatment were used. Samples were collected at 24 hr (Kitaake) or 48 hr (wildtype Komboka and edited lines) after inoculation. Total RNA was extracted using TRI reagent (TR 118; Euromedex, Souffelweyersheim, France). Following extraction, DNase I treatment was conducted using Turbo DNA-free kit (Thermo Fisher Scientific, AM1907). Subsequent synthesis of complementary DNA was carried out using SuperScriptIII (Thermo Fisher Scientific, 18080093) and oligo-dT primers. Quantitative PCR reactions were performed with SYBR Mesa Blue qPCR Mastermix (RT-SY2X-XXX Eurogentec, Seraing, Belgium). Three technical replicates were prepared for each sample. Expression values were normalized by subtracting the values obtained for reference gene *EF-1α* (GenBank: GQ848072.1) to the studied replicates ($2^{-\Delta Ct}$ method). Mean of the technical replicates for each sample was calculated. Three independent biological replicates were analyzed. Primer sequences are listed in *Supplementary file 1l*.

## Analysis of EBEs in *SWEET* promoters of *O. sativa* cv. Komboka

Genomic DNA was extracted from Komboka leaves using the CTAB method (*Li et al., 2013*). The gDNA fragments corresponding to promoters and first exons of *SWEET11a*, *13*, *14* were PCR-amplified using specific primer pairs. Amplicons were gel-purified and cloned into pJET1.2 (Thermo Fischer Scientific). Competent *E. coli* TOP10 competent cells were used for transformation (One Shot TOP10 Chemically Competent *E. coli*, Invitrogen, C404006). Plasmids were extracted from three individual colonies and were sent for Sanger sequencing (NucleoSpin Plasmid Mini Kit for plasmid DNA, Machinery-Nagel, 740588.50). Sanger sequencing reads representing promoter regions of *SWEET11a*, *13*, and *14* from Komboka were aligned with the respective promoter regions of Kitaake (Kitaake_OsativaKitaake_499. genome; Phytozome v.13) (*Supplementary file 1m*).

## Analyses of *Xa1* and *Xa4* sequences from *O. sativa* cv. Komboka

Full-length *Xa1* and four fragments of *Xa4* were PCR-amplified from *O. sativa* cv. Komboka gDNA using TAKARA PrimeSTART GXL polymerase (Takara Bio, R050A) and primers. Amplicons corresponding to full-length *Xa1* were subjected to Sanger sequencing. *Xa4* fragments were subcloned into pUC57Gent vector (Addgene #54338; *Binder et al., 2014*) and sequenced using Sanger sequencing. Individual sequence reads were mapped to their references and assembled via Geneious Prime 2022.1.1. Neighbor-joining phylogenetic analyses were made using CLUSTAL Omega (Geneious Prime). Genomic sequences of non-Komboka derived *Xa1* and *Xa4* had previously been published (*Hu et al., 2017*; *Ji et al., 2020*). Sequences of *Xa1* and *Xa4* genes used for phylogenetic analysis are provided in *Supplementary file 1c*.

## Hybrid CRISPR/Cas9 and Cpf1 vector construction

A new hybrid CRISPR-Cas9/Cpf1 system was developed to edit a maximum of six different targets (*Figure 5—figure supplement 2*). The hybrid CRISPR-Cas9/Cpf1 combines a duplex Cas9 gRNA combined with a multiplex Cpf1 cRNA. To develop this hybrid CRISPR-Cas9/Cpf1 system, we subcloned a human-codon-optimized *LbCpf1*-coding sequence (https://www.addgene.org/69988/sequences; *Zetsche et al., 2015*) and the rice Ubiquitin 1 (*OsUbi*) (LOC_Os02g06640) to create a gateway system pDEST:Cpf1. A PCR fragment of a Cas9 expression cassette, which contains a rice-codon-optimized *SpCas9*-coding sequence driven by the maize Ubiquitin 1 (*ZmUbi*) promoter (*Char et al., 2017*; *Zhou et al., 2014*), was amplified and inserted into pDEST:Cpf1 by Gibson Assembly (Gibson Assembly Master Mix, New England BioLabs, E2611S). Finally, we constructed the hybrid destination vector pDEST:Cpf1&Cas9. A Ribozyme-gRNA-Ribozyme (RGR) system was used to generate multiple cRNAs with different target sequences by flanking the cRNAs with a Hammerhead (HH) type ribozyme and a Hepatitis Delta Virus (HDV) ribozyme (*Gao and Zhao, 2014*). Guide RNA sequences are provided in *Supplementary file 1f*. The promoter of the small nuclear RNA gene from rice *OsU6.1* and *OsU3*, wheat *TaU3* and maize *ZmU3* were used to drive expression of RGG-cRNAs units. Four gBlock fragments synthesized by IDT (Integrated DNA Technologies, Inc, Iowa, USA) were inserted into the vector pTLN using *Xba*I and *Xho*I to produce four intermediate pUNIT vectors: pTL-OsU6.1_CpfRNA1, pTL-OsU3_CpfRNA2, pTL-TaU3_CpfRNA3, pTL-ZmU3_CpfRNA4. A double-stranded DNA oligonucleotide for each site was produced by annealing two complementary oligonucleotides. The

DNA sequence of positive clones was confirmed by Sanger sequencing. All four pUNIT vectors were transferred into another donor vector named pENTR4-ccdB using Golden Gate cloning (Golden Gate Assembly Kit, New England BioLabs, E1602S) to generate pENTR:cRNAs containing Cpf1 cRNAs cXo1, cXo2c, cTalF, and cXo2d (*Supplementary file 1f*). Simultaneously, a gTalC double-stranded oligonucleotide was inserted into the *BtgZ*I-digested pENTR-gRNAs (*Char et al., 2017*). The gRNA expression cassette was PCR-amplified with oligos U6P-F3 and -R3 inserted into pENTR:gRNAs at *Xba*I site through Gibson cloning to generate the hybrid donor plasmid pENTR:gRNA-cRNAs. The gRNA-cRNA cassette was mobilized to the hybrid destination vector pDEST:Cpf1&Cas9 using Gateway LR Clonase II Enzyme-Mix (Thermo Fisher Scientific, 11791020) to produce a pCam1300-CRISPR plasmid named pCpf1&Cas9:gRNAs-cRNAs (pMUGW5). DNA sequencing of the plasmid pMUGW5 detected the insertion of an *E. coli* IS element in the vector backbone. The insertion was already present in the original pCambia3000 in the BY stock collection and apparently did not affect transformation or editing.

## Rice transformation

*Agrobacterium*-mediated transformation of Komboka using immature embryos was performed as described (*Luu et al., 2020*). Briefly, the *Agrobacterium tumefaciens* strain LBA4404 was transformed with pMUGW5 via electroporation. Immature rice seeds at the late milky stage were harvested for immature embryo isolation. Immature embryos were inoculated with 5 µl *Agrobacterium* suspension (OD$_{600}$ 0.3) and incubated in the dark for 1 week. The emerged shoots were removed, and immature embryos were incubated for 5 days at 28 °C under continuous light. To select positive transformants, four rounds of hygromycin selection were applied (Hygromycin B solution, Carl Roth, CP 12.2), with each round consisting of 10 days. Resistant calli were moved onto a pre-regeneration medium and incubated for 10 days. Greening calli were transferred to a regeneration medium to develop small rice plantlets. Once plantlets reached 15 cm in height, they were transferred to soil and placed in a greenhouse. After 5 months, T1 seeds were harvested.

## Screening of EBE edited lines

To test whether generated T0 plants contained T-DNA insertion, the presence of *SpCas9*, *LbCpf1* and hygromycin resistance (*Hpt*) genes were checked. Leaf fragments (3–4 cm) were harvested for DNA extraction using a modified CTAB method for high-throughput DNA extraction (CTAB, Carl Roth, 9161.1). PCR was performed using GoTaq DNA Polymerase (GoTaq G2 Green Master Mix, Promega, M7823) with a melting temperature of 55 °C for *SpCas9*, *LbCpf1* and *Hpt*. For genotyping of EBE mutations, the four regions containing the six EBEs within the *SWEET11a*, *13* and *14* promoters were amplified with specific primers using Phusion HF polymerase (Thermo Fisher Scientific, F530S), and PCR amplicons were sequenced by Sanger sequencing (Microsynth seqlab). Chromatograms were analyzed manually using Benchling (https://www.benchling.com) to detect the mutations. Two rounds of transformation were performed. In Round 1, seven T0 plants were obtained from a single embryo, and all T0 plants contained biallelic mutations at all target sites (*Supplementary file 1g-i*). In the transformation Round 2, 18 T0 plants were T-DNA positive, from 18 independent immature embryos. Nine T0 plants were characterized further and found to contain biallelic mutations at all targeted sites (*Supplementary file 1g-i*). We screened 52 T1 plants from three independent events (#12, 14 and 16) and obtained seven lines with homozygous mutations in all six EBEs (*Supplementary file 1j-k*).

## Statistical analyses

Statistical analyses and graphical representations were prepared using R 4.0.5, on RStudio for Windows. Statistics were calculated using rstatix (https://cran.r-project.org/web/packages/rstatix). To determine if there was an effect of the treatment or the genotype on gene expression or lesion length, the medians of the different subsamples were compared using rstatix Kruskal-Wallis test (p-values <0.05). Then, mean groups were attributed to the different genotypes or treatments independently using rstatix Dunett's test (p-values <0.05). Letters were attributed to each mean group. Graphical representations were realized using package ggplot2 (https://ggplot2.tidyverse.org). Data were drawn as boxplots, delimited by the first and the third quartile of the distribution of the studied variable. The line inside the boxplot represents the median. The two lines that start from the boxplot join the minimum and maximum theoretical values. Outliers (<7% of total values)

were represented as black dots. The total number (n) of replicates for each strain/genotype condition is displayed next to each boxplot. Each observation is represented by a dot, with a color code by strain or genotype. The letters above each boxplot represent the mean groups calculated using Dunnett's test.

## Acknowledgements

We thank Paula Emmerich Maldonado (HHU) and Britta Killing (HHU) for excellent technical assistance. We thank Dr. Marietta Wolter for advice on using the QX200 Droplet Digital PCR (ddPCR) System (Biorad) at the Institute for Neuropathology, University Hospital Düsseldorf. The authors acknowledge the ISO 9001-certified IRD i-Trop HPC (South Green Platform) at IRD Montpellier for providing high-performance computing resources. WBF and YA acknowledge support by the Alexander von Humboldt Foundation. This work was made possible by funding from the Bill and Melinda Gates Foundation to Heinrich Heine University, Düsseldorf, with subawards to Iowa State University/University of Missouri, University of Florida, Institut de Recherche pour le Développement, and the International Rice Research Institute (OPP1155704). Research in the Frommer team was also supported by the Deutsche Forschungsgemeinschaft (DFG, German Research Foundation) under Germany's Excellence Strategy – EXC-2048/1 – project ID 390686111, and the Alexander von Humboldt Professorship, to WBF.

## Additional information

### Competing interests

The authors declare that no competing interests exist.

### Funding

| Funder | Grant reference number | Author |
| --- | --- | --- |
| Bill and Melinda Gates Foundation | OPP1155704 | Wolf B Frommer |
| Alexander von Humboldt-Stiftung | Professorship | Wolf B Frommer |
| Deutsche Forschungsgemeinschaft | EXC-2048/1 - project ID 390686111 | Wolf B Frommer |

The funders had no role in study design, data collection and interpretation, or the decision to submit the work for publication.

### Author contributions

Van Schepler-Luu, Supervision, Validation, Investigation, Methodology, Writing – original draft; Coline Sciallano, Investigation, Methodology, Writing – original draft; Melissa Stiebner, Formal analysis, Validation, Investigation, Visualization, Methodology; Chonghui Ji, Investigation, Methodology; Gabriel Boulard, Si Nian Char, Formal analysis, Investigation, Methodology; Amadou Diallo, Florence Auguy, Atugonza L Bilaro, Investigation; Yugander Arra, Kyrylo Schenstnyi, Formal analysis, Validation, Investigation, Methodology; Marcel Buchholzer, Formal analysis, Writing – original draft, Project administration; Eliza PI Loo, Formal analysis, Supervision, Validation, Writing – original draft; David Lihepanyama, Mohammed Mkuya, Resources, Investigation; Rosemary Murori, Resources, Data curation, Investigation, Methodology; Ricardo Oliva, Conceptualization, Investigation; Sebastien Cunnac, Conceptualization, Resources, Data curation, Formal analysis, Supervision, Validation, Investigation, Methodology, Writing – review and editing; Bing Yang, Boris Szurek, Conceptualization, Resources, Data curation, Formal analysis, Supervision, Funding acquisition, Validation, Investigation, Visualization, Methodology, Writing – original draft, Project administration, Writing – review and editing; Wolf B Frommer, Conceptualization, Data curation, Formal analysis, Supervision, Funding acquisition, Investigation, Visualization, Methodology, Writing – original draft, Project administration, Writing – review and editing

## Author ORCIDs

Van Schepler-Luu http://orcid.org/0000-0002-0709-2783
Coline Sciallano https://orcid.org/0000-0001-8988-7733
Melissa Stiebner https://orcid.org/0000-0001-8626-0951
Si Nian Char https://orcid.org/0000-0002-5759-0764
Yugander Arra https://orcid.org/0000-0001-6778-4258
Kyrylo Schenstnyi https://orcid.org/0000-0003-1595-6382
Marcel Buchholzer https://orcid.org/0000-0001-7485-6918
Eliza PI Loo http://orcid.org/0000-0002-7622-1139
Bing Yang http://orcid.org/0000-0002-2293-3384
Boris Szurek http://orcid.org/0000-0002-1808-7082
Wolf B Frommer http://orcid.org/0000-0001-6465-0115

## Decision letter and Author response

Decision letter https://doi.org/10.7554/eLife.84864.sa1
Author response https://doi.org/10.7554/eLife.84864.sa2

## Additional files

### Supplementary files

Supplementary file 1. Supplementary file with additional data. (a) List of *Xoo* strains used in this study (b) Disease survey in multiple rice growing areas in Tanzania in 2022. (c) Sequences of *Xa1* and *Xa4* genes from *Oryza sativa* cv. Komboka. (d) Characteristics of the TALome of *Xanthomonas oryzae* pv. *oryzae* Tanzanian strains. (e) *OsSWEET* promoter sequences for the rice varieties Komboka and Kitaake. (f) Guide RNA sequences. (g) EBE sequences in select *OsSWEET* promoters of Komboka wild-type and CRISPR-edited lines. (h) List of SNPs and Indels in *OsSWEET* promoters after editing of the rice varieties of Komboka and Kitaake (i) List of T0 lines with biallelic mutations at all targeted EBEs. (j) Screening of the T1 generation for homozygous mutations in all six *SWEET* EBEs. (k) List of T1 lines with homozygous mutations of all targeted EBEs. n.d. not determined. (l) List of primers used in this study (m) Alignment of *OsSWEET11a* promoters from the rice cv. Komboka and Kitaake.

Supplementary file 2. Accession numbers and main features of genome assemblies used for strain phylogeny inference.

Supplementary file 3. SWEET11a-targeting RVD sequences of the iTz strains.

Source data 1. Summary of raw data files available at dryad https://doi.org/10.5061/dryad.xpnvx0kk3.

MDAR checklist

### Data availability

All data supporting the results are available in the main text or supplementary materials. All data that support the findings of this study were included in the manuscript; raw data are available at Dryad (https://doi.org/10.5061/dryad.xpnvx0kk3; Summary of raw data files deposited at dryad is provided in Source Data 1). Sequencing data for strains from this study have been deposited in the NCBI Sequence Read Archive (SRA) database (Accession codes for iTz strains are provided in Supplementary File 2 - Tabs 1 and 2). Materials will be made available under MTA.

The following dataset was generated:

| Author(s) | Year | Dataset title | Dataset URL | Database and Identifier |
|---|---|---|---|---|
| Frommer WB, Szurek B, Yang B, Stiebner M, Schepler-Luu V, Scalliano C, Cunnac S, Auguy F, Schenstniy K | 2023 | Data for: Genome editing of an African elite rice variety confers resistance against endemic and emerging Xanthomonas oryzae pv. oryzae strains | https://doi.org/10.5061/dryad.xpnvx0kk3 | Dryad Digital Repository, 10.5061/dryad.xpnvx0kk3 |

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
