## [Editor Report]

This valuable study shows that new, virulent genotypes of Xanthomonas oryze pv. oryzae, that are similar to strains present in east Asia, cause outbreaks of bacterial blight of rice in Tanzania. The authors' use of CRISPR-based gene editing on multiple pathogen targets in an elite African rice variety to create lines resistant to both endemic and emerging pathogen strains in Africa makes for a compelling contribution to meet this alarming development.

---

## [Decision Letter]

**Decision letter after peer review:**

[Editors’ note: the authors submitted for reconsideration following the decision after peer review. What follows is the decision letter after the first round of review.]

Thank you for submitting the paper "Genome editing of an African elite rice variety confers resistance against endemic and emerging Xanthomonas oryzae pv. oryzae strains" for consideration by *eLife*. Your article has been reviewed by 3 peer reviewers, and the evaluation has been overseen by me as Reviewing Editor and Senior Editor. The reviewers have opted to remain anonymous.

Comments to the Authors:

After consultation with the reviewers, we have decided that this work cannot be considered in its current form for publication by *eLife*. However, we are interested in the work in principle and would reconsider a new version with additional key evidence, as outlined below. I very much hope that it will be possible to generate such limited additional data.

Since the advances presented here are tweaks, albeit important ones, to your own excellent prior work, the manuscript has to stand on a very clear description of the pathogens and a clear demonstration of the relevance of the edited lines in the field.

The overall framework of the manuscript, from outbreak analysis to genome-edited lines, is exemplary, but there are important questions as to the focus on a variety that is not susceptible under field conditions and also as to the inconsistent choice of pathogen strains for genome analysis and pathogenicity assays.

The inconsistency of pathogen data should hopefully be quite easily addressable by the inclusion of appropriate genomics analyses and pathogenicity assays. On the other hand, the fact that the rice variety chosen for editing is susceptible in artificial inoculation assays but not under field conditions tells us that there is more to learn on the fitness penalty that you may have missed. This brings the question of the immediate utility of these edited rice lines: The data presented here point to the need for edited lines not in the moderately resistant Komboka variety (what you have chosen), but in the varieties that are in fact susceptible to the emerging strains under field conditions.

I would happy to hear if you have justification for this, and could make an argument for this – what seemed to us – peculiar choice.

*Reviewer #1 (Recommendations for the authors):*

1. Citing Supplementary table 1 for the sentence on the characterization including diagnostics and pathogenicity of 833 strains might be misleading. This table contains only a subset of strains and their NCBI accession numbers. I see some publications cited here and it may be that some of these strains are part of those publications. So, it might be misleading to the readers to refer them to supplementary table 1 for the collection.

2. "We performed disease surveys in this area in 2019 and 2021 and identified two outbreaks on TXD 306 (SARO-5), a rice variety popular in irrigated ecologies in Tanzania. More recently BB was identified in Komboka. Leaves with typical BB symptoms were processed, and seven strains from Dakawa and 106 from Lukenge were isolated and validated as Xoo with diagnostic primers (Lang et al., 2010). Notably, surveys performed in subsequent years showed increased severity and spread (Supplementary Table 2)." The way this writeup is, it seems like the increased severity and spread was seen on Komboka. It might be good to clarify that disease severity of 2 (4-6%) of disease was observed on Komboka. But other varieties showed more disease severity of scales 4 or 5.

3. Check spelling in Figure for Teqing. Figure 1 supplement 2. Is it Xa4_Teping or Xa4_Teqing?

4. It might be good to include TALes in endemic and emerging strains in the table with a list of strains and their origin. Similarly, resistance gene profiles for African rice varieties sampled in surveys can be included (based on the database). To Xanthomonas oryzae readers, this may be obvious. However, to the scientific community interested in outbreak analysis and management, this study could be an excellent example to refer to.

*Reviewer #2 (Recommendations for the authors):*

1. The work could be strengthened with additional draft genome sequences, which may confirm that the outbreak is genetically homogenous and likely the result of a single introduction. Annotating TALes in all three sequenced strains would show if the three strains have the same effector profile. It would be helpful to include metadata in Table S1 to help the reader understand which strains were used for which experiments, and which disease outbreak they are associated with.

2. It would be helpful if outbreak data were summarized and the relationships of the various outbreaks to each other and to the work presented in this manuscript were made clearer. For example, if data were summarized and placed on a map for readers not familiar with the regions and districts of Tanzania.

On page 4, when describing the outbreaks of BB, it's unclear which outbreaks are first reported in the literature in this study versus what has been previously reported. It's not clear how supplementary table 1 relates to the rest of this sentence: "To systematically analyze these newly isolated strains and to compare them to the broader African Xoo landscape, 833 strains from rice fields sampled in nine African countries between 2003 and 2021 were collected and characterized using molecular diagnostic as well as pathogenicity assays (Supplementary Table 1)(Afolabi et al., 2016; Gonzalez et al., 2007; Tall et al., 2020; Tekete et al., 2020). Supplementary Table 2 is cited for "two unprecedented outbreaks were identified, one in 2019 in Dakawa and another in 2022 in Lukenge" but this table only includes 2022 data. The sentence "We performed disease surveys in this area in 2019 and 2021 and identified two outbreaks on TXD 306 (SARO-5), a rice variety popular in irrigated ecologies in Tanzania.", has no supporting data, except that strains from 2019 are used. Finally, the sentence "More recently BB was identified on Komboka" is shown in Supplementary Table 2 after many pages of other reports, but the table isn't cited. Improving the clarity of presentation of these reports would be beneficial to the reader so that there isn't a need to synthesize data among text and supplementary tables to understand which outbreaks were sampled and how they are used in this study.

3. On page 5, the text says "Notably, CIX4457, CIX4458, and CIX4462 clustered with Asian Xoo isolates (Figure 2A)." But only 4462 is shown in the tree, the other strains shown are 4506 and 4509.

4. There are two hyperlinks at the top of page 4 that are not working.

*Reviewer #3 (Recommendations for the authors):*

The work in the manuscript under review is technically quite sound. However, the description of the work could be improved. For example, the first paragraphs of the Results section describe exclusively published and not new data. I suggest moving these passages to the Introduction section. Moreover, the manuscript has several tables and figures that are referenced but hardly described in the Results section. I would suggest that the authors provide at least a basic description of the contents of these Tables/Figures in the Results text. One cannot expect that the reader will go through the data to extract the essential information.

---

## [Author Response]

[Editors’ note: The authors appealed the original decision. What follows is the authors’ response to the first round of review.]

Comments to the Authors:After consultation with the reviewers, we have decided that this work cannot be considered in its current form for publication by eLife. However, we are interested in the work in principle and would reconsider a new version with additional key evidence, as outlined below. I very much hope that it will be possible to generate such limited additional data.Since the advances presented here are tweaks, albeit important ones, to your own excellent prior work, the manuscript has to stand on a very clear description of the pathogens and a clear demonstration of the relevance of the edited lines in the field.The overall framework of the manuscript, from outbreak analysis to genome-edited lines, is exemplary, but there are important questions as to the focus on a variety that is not susceptible under field conditions and also as to the inconsistent choice of pathogen strains for genome analysis and pathogenicity assays.The inconsistency of pathogen data should hopefully be quite easily addressable by the inclusion of appropriate genomics analyses and pathogenicity assays. On the other hand, the fact that the rice variety chosen for editing is susceptible in artificial inoculation assays but not under field conditions tells us that there is more to learn on the fitness penalty that you may have missed. This brings the question of the immediate utility of these edited rice lines: The data presented here point to the need for edited lines not in the moderately resistant Komboka variety (what you have chosen), but in the varieties that are in fact susceptible to the emerging strains under field conditions.I would happy to hear if you have justification for this, and could make an argument for this – what seemed to us – peculiar choice.

We are happy to see the comments of the editor on the value and importance of this work. We substantially edited the text, added a map as new Figure 1, included new data on the close relatedness of the strains from Dakawa and Lukenge (replaced Figure 3), added a new Figure 2 Supplement that shows that besides the Dakawa strains, also a Lukenge strain is fully virulent on Komboka, and we added sections to Discussion and Methods providing more depth regarding the predictive power of the greenhouse clipping assay for field performance with references.

Some key points that we think are relevant for the interpretations made by the reviewers:

Until now, virulence of *Xoo* depends fully on the ability to induce one of the three *SWEET* genes *SWEET11a, 13 or 14*.To our knowledge, the systematic editing of SWEET EBEs is the only effective way to obtain resistance against a large panel of Asian and African *Xoo* strains (many hundreds were tested by us).Indian breeders regularly test a panel of NIL rice lines carrying either single or combinations of R/genes against a panel of strains.To date, All single *R* genes have been broken in India.Similarly, all double, triple and quadruple *R* gene combinations have been broken.At present, Indian breeders have to combine at least five *R* loci (often genes or mechanisms are not known) to obtain resistance to the majority, but not all, strains.Thus, even if a variety may be resistant to a particular strain, or carries a combination of four *R* genes, it makes sense to edit the EBEs of all *SWEETs* in this variety to provide robustness.Komboka was chosen carefully after consulting with the key breeders in order to use the most advanced line that has a high chance of being adopted and that will be widely used at the time it can be released, which is many years only after it was successfully generated.Note also, that the top BB experts and breeders confirmed that leaf clipping assays in greenhouses are predictive for field performance of resistance (actually even moderate resistance in the clipping assay will behave as resistant in the field).Komboka is fully susceptible to the strains from the Tanzanian outbreak in clipping assays.The data on Komboka from the 2022 survey are first observations, and cannot be used to deduce the level of resistance in the field. This is particularly due to difficulties in assessing disease incidence in the field in such a survey: infection may have occurred early or late, with high titer, or observation may have been made late, when plants start to senesce, and thus disease could not be adequately classified. Note that due to the recent detection of the outbreak, systematic multi-year observations have only been initiated.Note also, that we believe it will be important to generate such edits in a wide range of varieties, since there is a lot of consumer preference in different countries and regions. Based on the large combinatorial space (if we make a low assumption, namely that only the first 15 bp are relevant, and we assume that each repeat can recognize 2 different bases, we can generate 2^15^ variants), we would recommend to deploy different edits in different varieties to increase the robustness.

You asked us to justify the use of Komboka. This request was likely based on a comment from Reviewer 1, who stated that Komboka is moderately resistant.

The Reviewer bases his comment on Komboka susceptibility from our newest data from 2022 (!) from a single observation in one location. It is for obvious reasons not possible to conclude anything about the grade of susceptibility from a single observation. One needs to observe over several years and several sites. This is the first observation in the field for Komboka and it could be more severe in 2023. Komboka was originally described on websites as moderately resistant to BB. However, we do not believe that such a classification is relevant. It depends on the particular strain and the gene-for-gene interaction as well as the *R* gene outfit of the rice line whether the rice variety in this particular pair can be classified as susceptible or resistant, or the strain as virulent or avirulent. If you look at Figure 2, it is clear that Komboka is fully susceptible to PXO86 and the three strains isolated in 2019 in Dakawa-Tanzania, but resistant to a wide range of African strains from our collection. Note that these strains were chosen for the initial characterization of Komboka to evaluate it’s resistance/susceptibility profile. We then tried to find out why Komboka is resistant to a lot of the African stains, and found that it carries *Xa1* and *Xa4*. Subsequent analyses showed that Komboka is also susceptible to PXO99^A^ and to the Lukenge strains, which were isolated in Tanzania in 2021; based on genotyping they are close relatives and possibly descendants or the strains first strains found in Dakawa, although we cannot rule out the reverse. In summary, if iTz strains ente Kenya (which we hypothesize may already happened), they will cause severe damage, faster manifestation. The edited lines will be the current best tools to help the farmers there.The key assay to evaluate field-relevant resistance that is used by breeders is ‘clip infection’ (Kauffman et al., 1973). This is an ‘overkill’ assay (OD600 = 0.5, so billions of bacteria), that is, according to all breeders we have talked to, e.g., in India, as well as several senior *Xoo* experts, highly predictive: they are **not** aware of any cases where a line tested with this assay and classified as resistant in greenhouses or growth chambers did not hold up in the field. It is generally even assumed that moderate resistance, so some lesion detected in this assay, will be fully resistant in the field. We added text to Results and Discussion sections, and a new section on sampling in Methods with respective references that show the correlation of data using assays with the same strains in greenhouse and field.Different countries have a preference for growing certain rice varieties. Since Kenya was projected to be among the first African countries to develop editing guidelines, we evaluated germplasm from Kenya. Thus, instead of using Kitaake as proof of concept, or IR64, we focused on the promising elite variety Komboka for Kenya as recommended by IRRI Africa.We selected Komboka in 2019 as a candidate because it was recommended as an emerging variety in Kenya with exceptionally high yield, and suitable properties relevant to the local consumers. This was a bet at that time, but the variety has indeed been widely adopted in Kenya (pers. Comm. IRRI Africa). The next step after acquiring seeds was to establish whether these can be grown and be fertile in Düsseldorf in the greenhouse (tropical variety that normally prefers neutral or short day conditions). We successfully established conditions for effective growth and reproduction in Düsseldorf and growth conditions in Montpellier for virulence tests, then needed to establish a transformation protocol, which was published recently (*Luu et al. 10.21769/BioProtoc.3739*).In India, most single, double and triple *R* gene combinations for BB have been broken. Thus, it is necessary, even if a variety is resistant to a particular strain, to generate robust broad spectrum resistance. Since virulence of *Xoo* depends critically on *SWEET* activation (no one has yet observed a case where *SWEET11a*, *13* or *14* activation was not involved – note the single manuscript that stated that has been retracted), and US strains are not virulent because they do not have TAL effectors. We thus used knowledge of the TALe repertoire of our worldwide collection to edit all known *SWEET* EBEs. These lines are resistant representatives trains from Asia and Africa, but most importantly, to the strains from the outbreak described here.We have clarify the disease spread from the outbreak, both by editing the text, as well as by adding a new Figure 1 with maps of the collection sites and % infected plants in the respective sampling years. We provide more information on the surveys and the sampling. We also clarified the nomenclature of the introduced strains and provide more data on their relationship (see also next paragraph).Nomenclature issues: We obviously tried something that was not very wise, we used a code (CIX*nnnn*) to systematically number the strains that were collected. This is beneficial, especially since before strains obtained random labels such as ‘*Nati Park’*. But we understand that this code can be confusing since it is hard to remember the similar strain numbers. We have thus, while keeping the code, given logical names to the isolates – *i* for the newly introduced Asian strains, Tz for collected in Tanzania (we have started to collect in Kenya and will expand widely, so need Tz in there), Dak for Dakawa and *Luk* for Lukenge, and the year of isolation (21 for 2021), plus a number for the isolate (e.g.: iTzLuk21-3). This makes it very intuitive to trace the strains used in the various experiments in publications. This should address a number of the issues brought up by the reviewers. It is also noteworthy, that the project had two parallel activties - selection of a suitable rice line (see below) for Kenya, first virulence tests with available strains, followed by characterization of the *R* gene outfit followed by editing. In parallel, we collected strains in Tanzania before and after the outbreak, but the iTz were not available at the beginning. We apologize for the confusion the nomenclature may have caused.

Reviewer #1 (Recommendations for the authors):1. Citing Supplementary table 1 for the sentence on the characterization including diagnostics and pathogenicity of 833 strains might be misleading. This table contains only a subset of strains and their NCBI accession numbers. I see some publications cited here and it may be that some of these strains are part of those publications. So, it might be misleading to the readers to refer them to supplementary table 1 for the collection.

We apologize, there was indeed a disconnect. We chose 21 representative strains based on the country of origin, knowing that the genotyping of our collection of 833 strains from across Africa shows that Africain Xoo strains tend to cluster according to the geography. This is another large study that is currently being written up. We clarified this in the new version of the manuscript.

2. "We performed disease surveys in Morogoro in 2019 and 2021 and identified two outbreaks on TXD 306 (SARO-5), a rice variety popular in irrigated ecologies in Tanzania. More recently BB was identified in Komboka. Leaves with typical BB symptoms were processed, and seven strains from Dakawa and 106 from Lukenge were isolated and validated as Xoo with diagnostic primers (Lang et al., 2010). Notably, surveys performed in subsequent years showed increased severity and spread (Supplementary Table 2).

We tried to clarify the information from former Table S1 (now S2) by adding maps as the new Figure 1. We have now highlighted that these are snapshots, and infection of Komboka was detected only recently in one location (most recent collection in 2022). Komboka is fully susceptible (see clipping assay results in the various figures) to the introduced Tanzanian strains. Note the ‘overkill” nature of the clipping assay which by all experts we talked to is highly predictive for field performance. Note also that disease severity is affected by many factors such as climate conditions, nitrogen fertilizer amount, initial infection titer and path and time of infection relative to plant growth stage. Also, the observations were made at one time point, and thus could either represent an early time point after infection. It seems likely that the strains spread recently, and the disease has not fully manifested. Future surveys will clarify this. But again, the breeders and experts we talked to confirmed that the clipping assay is highly sensitive, thus our greenhouse observation that Komboka shows lesion lengths similar to PXO99^A^ and in the range of 15-20 cm is consistent with full susceptibility to the iTz strains. Please also see comments to editor and to comment #1 of this reviewer above. We clarified this also in the text. We added text to Results and Discussion sections and a new section on sampling in Methods with respective references that show the correlation of data from assays with the same strains in greenhouse and field.

3. Check spelling in Figure for Teqing. Figure 1 supplement 2. Is it Xa4_Teping or Xa4_Teqing?

We apologize the typo, which has been corrected.

4. It might be good to include TALes in endemic and emerging strains in the table with a list of strains and their origin. Similarly, resistance gene profiles for African rice varieties sampled in surveys can be included (based on the database). To Xanthomonas oryzae readers, this may be obvious. However, to the scientific community interested in outbreak analysis and management, this study could be an excellent example to refer to.

We are aware that it is very hard for non-experts to follow the nomenclature of strains and TALes. We have followed the suggestion, and where available, added the TALe information for all strains used in the new Supplementary Table 1 and 3 which summarizes info on the strains used here.

Reviewer #2 (Recommendations for the authors):1. The work could be strengthened with additional draft genome sequences, which may confirm that the outbreak is genetically homogenous and likely the result of a single introduction. Annotating TALes in all three sequenced strains would show if the three strains have the same effector profile. It would be helpful to include metadata in Table S1 to help the reader understand which strains were used for which experiments, and which disease outbreak they are associated with.

As indicated above, we have provided new data resulting of the analysis of the draft genome sequences of 5 additional strains from Dakawa and Lukenge (new Figure 3), demonstrating that strains from Dakawa and Lukenge all share the same typical Asian-like virulence features: presence of *pthXo1B* and *iTALes*. Moreover, they all cluster together as one monophyletic group upon SNP-based whole genome sequence phylogeny, therefore further confirming the homogeneity of the outbreak. We have added a new Suppl. Table 3 that provides all these information. As to the 3 strains for which the chromosome could be circularized and the respective complete TALe repertoires extracted, we have provided, as requested by reviewer #2, a new Suppl. Table (Suppl. Data 1) displaying the alignments of the RVD arrays of 17 TALE groups. As expected, the 3 strains show the same effector profile.

2. It would be helpful if outbreak data were summarized and the relationships of the various outbreaks to each other and to the work presented in this manuscript were made clearer. For example, if data were summarized and placed on a map for readers not familiar with the regions and districts of Tanzania.

Thank you for the suggestion. We have followed the advice and added a new Figure 1 with the respective maps as well as photos of the infection as supplementary Figures.

On page 4, when describing the outbreaks of BB, it's unclear which outbreaks are first reported in the literature in this study versus what has been previously reported. It's not clear how supplementary table 1 relates to the rest of this sentence: "To systematically analyze these newly isolated strains and to compare them to the broader African Xoo landscape, 833 strains from rice fields sampled in nine African countries between 2003 and 2021 were collected and characterized using molecular diagnostic as well as pathogenicity assays ( Supplementary Table 1)(Afolabi et al., 2016; Gonzalez et al., 2007; Tall et al., 2020; Tekete et al., 2020).

Thanks for pointing that out. We moved this part of the Results to where it belongs – the introduction. Thus, the *Results section* now describe our results. We also admit that it was unclear why we mention the large African collection. We have collected these strains over a long period and are assembling a manuscript that describes the genetic diversity of this collection. From the collection of endemic African strains, we used a carefully chosen representative subset for testing virulence on Komboka. This has now been clarified in the first section of the Results.

Supplementary Table 2 is cited for "two unprecedented outbreaks were identified, one in 2019 in Dakawa and another in 2022 in Lukenge" but this table only includes 2022 data. The sentence "We performed disease surveys in this area in 2019 and 2021 and identified two outbreaks on TXD 306 (SARO-5), a rice variety popular in irrigated ecologies in Tanzania.", has no supporting data, except that strains from 2019 are used. Finally, the sentence "More recently BB was identified on Komboka" is shown in Supplementary Table 2 after many pages of other reports, but the table isn't cited. Improving the clarity of presentation of these reports would be beneficial to the reader so that there isn't a need to synthesize data among text and supplementary tables to understand which outbreaks were sampled and how they are used in this study.

Sorry, this was a misnumbering, which has been corrected. We have clarified the text and included data for the identification of BB in Dakawa in 2019 and Lukenge in 2021. We edited also the text regarding the survey and added new information for the survey performed in 2022.

3. On page 5, the text says "Notably, CIX4457, CIX4458, and CIX4462 clustered with Asian Xoo isolates (Figure 2A)." But only 4462 is shown in the tree, the other strains shown are 4506 and 4509.

Apologies for the confusion. We had initiated WGS of all strains collected, however, the quality of some of the strain sequences was not high enough to be included. We have now included data from 5 new draft genomes demonstrating that all the newly introduced strains form a genetically homogeneous group. We cannot determine which strain was at the origin of the outbreak, however it is clear that these strains are all highly related and derive from a close common ancestor (introduction). We also show that the strains from Dakawa and Lukenge contain *pthXo1*-like TAL effectors that target *SWEET11a*, as well as *iTALes*. We also changed the nomenclature since the CIX codes are highly valuable on the long run, e.g. in databases, but are hard to follow for the reader. The new names are intuitive and much easier to follow.

4. There are two hyperlinks at the top of page 4 that are not working.

We tested all hyperlinks, which are all functional.

Reviewer #3 (Recommendations for the authors):The work in the manuscript under review is technically quite sound. However, the description of the work could be improved. For example, the first paragraphs of the Results section describe exclusively published and not new data.

The Reviewer is perfectly right. We moved that section to the Introduction and revised the first part of the Results.

I suggest moving these passages to the Introduction section. Moreover, the manuscript has several tables and figures that are referenced but hardly described in the Results section. I would suggest that the authors provide at least a basic description of the contents of these Tables/Figures in the Results text. One cannot expect that the reader will go through the data to extract the essential information.

We now added sentences in Results to better include the supp data.